# LFA-1 interaction with GBP-130 on *Plasmodium falciparum*-infected red blood cells mediates NK cell activation and parasite control

Osama Mukhtar[1], Ravi Dutt[2], Ashutosh Panda[1], Poonam Kumari[1,3], Suneet Shekhar Singh[1], Gourab Paul[1,4], Neha Prakash[5], Madiha Abbas[5], Md Muzahidul Islam[5], Priya Arora[5], Alma Tammour[1], Asif Mohmmed[5], Dhiraj Kumar[2]*, Pawan Malhotra[1]*

[1]Malaria Biology Group, International Centre for Genetic Engineering and Biotechnology (ICGEB), Aruna Asaf Ali Marg, New Delhi, India; [2]Cellular Immunology Group, International Centre for Genetic Engineering and Biotechnology, New Delhi, India; [3]Amity Institute of Biotechnology (AIB), Amity University, Kant Kalwar, NH-11C, Jaipur, India; [4]Department of Bioinformatics, Amity School of Biological science, Amity University, Jaipur, India; [5]Parasite Cell Biology Group, International Centre for Genetic Engineering and Biotechnology, New Delhi, India

*For correspondence:
dhiraj@icgeb.res.in (DK);
pawanmal@gmail.com (PM)

## eLife Assessment

This **useful** study addresses the interesting question of how immune cells recognise infected erythrocytes in malaria. It proposes the parasite protein PfGBP-130 as an interaction partner of the human cell surface protein LFA 1, which could help explain how NK cells recognize infected erythrocytes. The conclusions are partially supported by pull-down and cell-based activation data. However, the overall evidence of direct interaction at the cell-cell interface and downstream effects is **incomplete**; stronger evidence is required to demonstrate surface exposure of PfGBP-130, as well as a direct role of this antigen in killing.

**Abstract** Natural killer (NK) cells contribute to early immunity against *Plasmodium falciparum* by recognizing and eliminating infected red blood cells (iRBCs), a process mediated in part by the integrin LFA-1. However, the cognate parasite ligand for LFA-1 has remained unknown. Here, we identify glycophorin binding protein-130 (*Pf*GBP-130) as a surface-expressed ligand on iRBCs that binds the I-domain of LFA-1 (LFA-1 αI). Using an LFA-1 αI-Fc fusion protein, we demonstrate stage-specific binding to iRBCs, and LC-MS/MS analysis of immunoprecipitates of αI-Fc bound to iRBC revealed *Pf*GBP-130 as a high-confidence interactor. Recombinant *Pf*GBP-130 binds NK and THP-1 cells in an LFA-1-dependent manner. Co-culture assays show that *Pf*GBP-130 promotes NK cell activation and degranulation and facilitates contact-dependent killing of iRBCs. Neutralizing antibodies against *Pf*GBP-130 significantly impair these responses. Our findings establish *Pf*GBP-130 as the LFA-1 ligand on iRBCs, providing new insight into NK cell-mediated immunity in malaria and identifying a potential target for host-directed interventions.

## Introduction

Natural killer (NK) cells are innate immune cells involved in early defense against microbial pathogens, parasitic infections, and tumor cells (*Burrack et al., 2019*). NK cells are found in various tissues, including the liver, peritoneal cavity, and placenta, and constitute up to 15% of peripheral blood lymphocytes. NK cells were initially identified due to their ability to kill certain tumor cells in vitro (*Cerwenka and Lanier, 2001*). Subsequent studies using in vivo murine models demonstrated that depletion of NK cells leads to enhanced tumor formation (*Kärre, 2002*). Activation of NK cells results from the concerted action/response of cytokine signaling, adhesion molecules, and the interaction of activating receptors with their corresponding ligands expressed on the surface of pathogenic infected cells or tumors. Despite having understood the role(s) of NK cells in controlling microbial infections and tumor surveillance, the mechanisms of NK cell recognition by pathogenic cells are not well understood.

Malaria infection and pathogenesis involve a complex series of interplays between parasite and human host factors, and a better understanding of host cells (factors) may be important in developing innovative host-derived therapeutic approaches. NK cells provide the first line of defense against malaria parasite infection. Studies in mouse malaria models have shown that various immune cells such as NK cells, dendritic cells, T cells, and B cells contribute to antiparasitic immunity (*Chen et al., 2014*; *Ye et al., 2018*). *Plasmodium* development in humans occurs in two phases: the merozoite stage in the liver and the symptomatic blood stage. During the liver stage, immunity against *Plasmodium* is primarily mediated by interferon-gamma (IFN-γ). NK cells secrete IFN-γ, which eliminates infected hepatocytes either directly or by activating other immune cells, including cytotoxic T cells and macrophages (*Artavanis-Tsakonas and Riley, 2002*; *Burrack et al., 2019*).

NK cells contribute to blood stage immunity through three main mechanisms. First, they facilitate the clearance of infected red blood cells (iRBCs) via a bystander mechanism, involving the secretion of IL-12 and IL-18. These cytokines prompt macrophages and dendritic cells to clear iRBCs or stimulate adaptive immune cells. Second, NK cells engage in antibody-dependent cell-mediated cytotoxicity (ADCC), wherein antibody-opsonized iRBCs are recognized via the CD16 (FcγRIII) receptor. This leads to the release of cytotoxic granules, including perforin and granulysin, enabling targeted lysis of infected cells. Third, NK cells can directly kill infected or transformed cells through activating receptor-ligand interactions, a mechanism well characterized in antiviral and tumor immunity. NK cells are critical in limiting acute malaria infection, as depletion of NK cells in mouse malaria models has been associated with higher parasitemia and accelerated disease progression (*Chen et al., 2014*; *Ye et al., 2018*). NK cell protective effects are mediated through both cytotoxic activity and secretion of IFN-γ. Clinical studies in *Plasmodium falciparum*-infected children have revealed an inverse correlation between NK cell frequency or functional activity and parasite burden (*Ojo-Amaize et al., 1981*). Furthermore, experimental infections in malaria-naïve individuals have demonstrated that NK cells are among the earliest responders, capable of directly lysing iRBCs and releasing IFN-γ and soluble granzyme (*Hermsen et al., 2003*; *Ye et al., 2018*).

*Plasmodium* iRBCs activate NK cells, enhancing IFN-γ production and cytotoxicity through perforin and granzyme release. This activation is contact-dependent, as demonstrated by the lack of upregulation of NK cell activation markers during transwell incubation. Co-incubation of iRBCs with NK cells promotes *Plasmodium* parasite killing, and this contact-dependent interaction is mediated in part by the lymphocyte function-associated antigen-1 (LFA-1) on NK cells (*Chen et al., 2014*; *Korbel et al., 2005*). In this study, we demonstrate that iRBCs bind specifically to the 'αI domain' of LFA-1 and identify *Plasmodium* glycophorin binding protein-130 (*Pf*GBP-130) as a major iRBC surface ligand interacting with LFA-1. Furthermore, we show that *Pf*GBP-130 expressed on the iRBC surface binds to THP-1 and NK cells, inducing activation and degranulation of these cells. Together, our findings identify *Pf*GBP-130 as a ligand for LFA-1 that facilitates direct contact-mediated killing of *Plasmodium*-infected erythrocytes by NK cells, highlighting its potential role in host-parasite immune interactions.

## Results

### LFA αI domain of LFA-1 is involved in ligand binding on iRBC surface

Contact-dependent killing of *Plasmodium* iRBCs by NK cells is essential for effective malaria control and is mediated by LFA-1 (CD11a/CD18), a β2 integrin composed of an αL (CD11a) and

β (CD18) subunit. Notably, CD11a contains an inserted (I) domain of approximately 200 amino acids, homologous to the von Willebrand factor type A (vWA) domain family (*Figure 1—figure supplement 1A*), which plays a critical role in ligand binding, including interaction with ICAM-1 (*Qu and Leahy, 1995*). Here, we generated an LFA-1 αI-Fc fusion protein to assess αI domain binding to iRBCs. Schematics of CD11a (a subunit of LFA-1) and its I domain fused to human IgG1 (hIgG1) Fc domain is shown in *Figure 1A*. DNA sequences corresponding to LFA-1 αI-Fc fusion protein were cloned in a plasmid pFUSE-hIgG1-Fc2 (*Figure 2—figure supplement 1A and B*), and the protein was expressed in Chinese hamster ovary (CHO) K1 cells under serum-free conditions. Fusion protein was purified via Protein A affinity beads, and purified protein was analyzed by SDS-PAGE and western blot using secondary anti-human IgG (hIgG). A protein of an ~45 kDa was seen in SDS-PAGE and western blot (*Figure 1B*), which corresponded to the molecular weight of fusion construct. We next analyzed the binding of purified LFA αI-Fc protein with iRBCs and detected binding using secondary human antibody conjugated to PE-Texas Red or anti-human antibody conjugated to FITC. Binding was also analyzed for all the three asexual blood stages of *Plasmodium* ring, trophozoites, and schizonts by FACS. As shown in *Figure 1C*, high-affinity binding was observed with all three stages and was significantly higher than that observed with the hIgG isotype control.

To exclude the possibility that LFA-1 αI-Fc binding to iRBCs resulted from non-specific interactions with erythrocyte surfaces rather than parasite-derived antigens, we performed parallel incubations using uninfected RBCs and an isotype-matched human IgG control. Little or no binding was observed for these molecules to the infected cells. No significant difference in binding to uninfected RBCs was observed between LFA-1 αI-Fc and the isotype control, indicating the absence of non-specific erythrocyte binding (*Figure 1—figure supplement 1B*). These results showed that the αI domain of LFA-1 is involved in binding to iRBCs.

## *P. falciparum* glycophorin binding protein-130 protein on iRBC surface binds to LFA αI domain of LFA-1 protein

To elucidate the molecular basis of LFA-1 mediated interactions with *P. falciparum*-infected erythrocytes, we undertook a comprehensive proteomic approach. Briefly, recombinant LFA-1 αI domain Fc fusion protein (LFA-1 αI-Fc) used as an affinity reagent to capture iRBC surface-interacting proteins following DTSSP cross-linking. hIgG1 served as a control to distinguish specific from non-specific background interactions. Immunoprecipitated proteins from both LFA-1 αI-Fc and hIgG1 conditions were analyzed by LC-MS/MS, enabling a comparative assessment of specifically interacting proteins. Three independent biological replicates were performed, and stringent selection criteria were applied such that only proteins supported by detection of ≥5 peptides were considered for downstream analysis. Although several proteins showed similar sequence coverage, these were also present in beads+human IgG control eluates, indicating non-specific background binding (*Supplementary file 1A*). Only proteins consistently detected across all replicates were considered. *Pf*GBP-130 (PF3D7_1016300) was identified as a top candidate based on peptide coverage and spectral abundance across all the replicates (*Figure 2A*). LC-MS/MS analysis thus strongly suggested a specific and robust interaction between LFA-1 αI domain and *Pf*GBP-130.

To validate the observed interaction and provide orthogonal confirmation of our proteomic findings, we expressed the N-terminal region of *Pf*GBP-130 (aa 69–270 encompassing one GBP repeat) in *E. coli* and purified the protein to near homogeneity (*Figure 2Bi*). Polyclonal antibodies were raised against recombinant GBP-130 N-terminal protein fragment referred to here as *Pf*GBP-130-N in rabbit (*Figure 2Bii*). Antibodies raised against *Pf*GBP-130-N were specific as they could detect the native *Pf*GBP-130 on iRBC surface. We carried out co-localization IFA with *Pf*GARP, a well-established iRBC surface protein with an extracellular domain (*Raj et al., 2020*). Anti-*Pf*GBP-130 N staining co-localized with *Pf*GARP at the iRBC surface, with a Pearson's correlation coefficient of 0.67, supporting surface exposure of *Pf*GBP-130 (*Figure 2Biii*). To confirm the specificity of anti-*Pf*GBP-130 N antibodies, we next performed a western blot analysis of trophozoite-stage parasite lysate and also for the immunoprecipitated fraction of LFA-1 αI-Fc protein-bound iRBC lysate using anti-*Pf*GBP-130 N antibodies. As shown in *Figure 2Biv*, a distinct band at ~95 kDa, consistent with the reported molecular weight of *Pf*GBP-130 (*Kochan et al., 1986*), was observed exclusively within the LFA-1 αI-Fc-bound iRBC eluate, as well as in iRBC lysate control. Critically, this band was absent in the hIgG1 control eluate, confirming

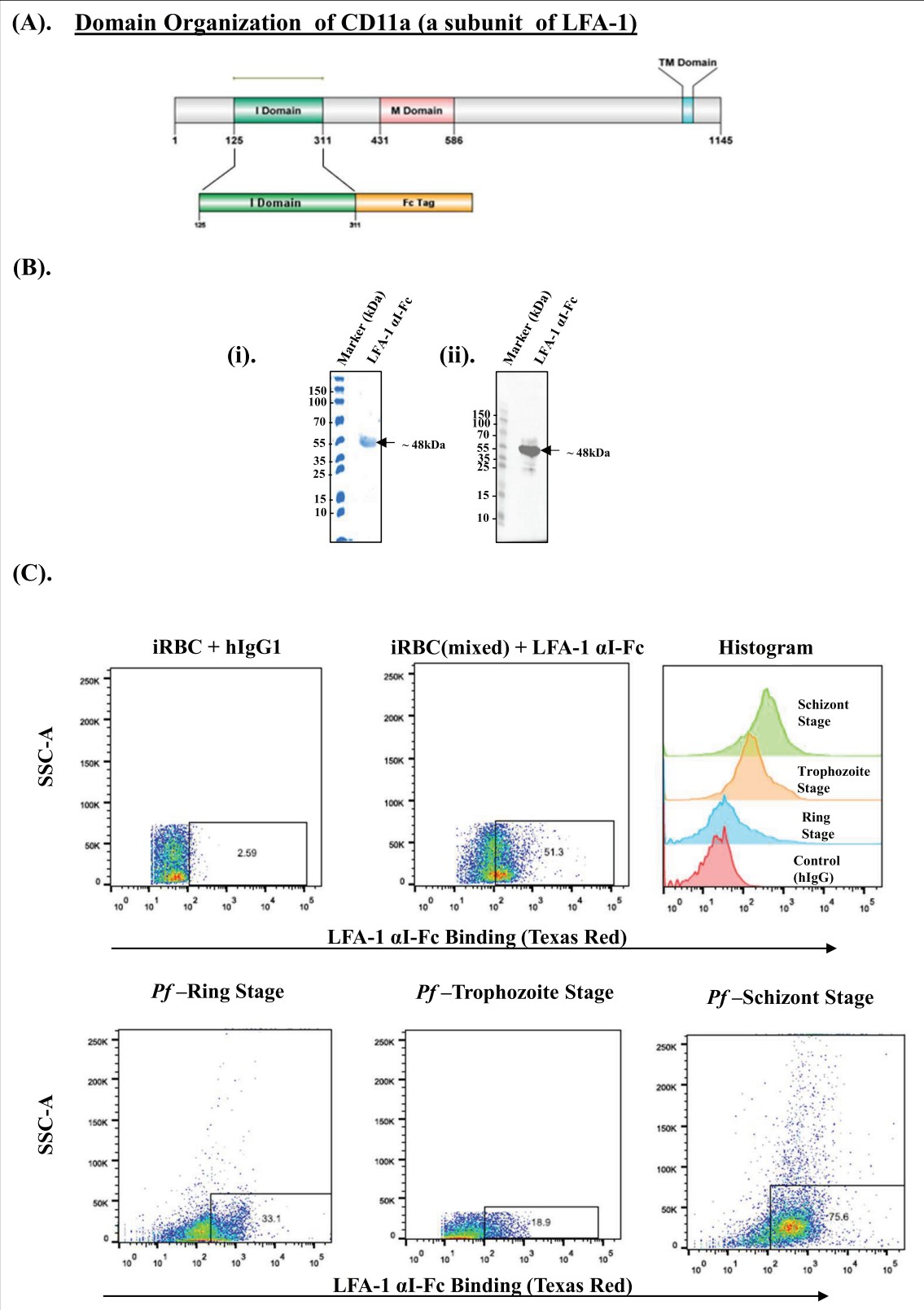

**Figure 1.** The αI domain of lymphocyte function-associated antigen-1 (LFA-1) binds to *P. falciparum*-infected erythrocytes. (**A**) Schematic representation of full-length CD11a, a subunit of LFA-1, and the recombinant construct encoding the C-terminal Fc-tagged αI domain of LFA-1. (**B**) Expression and purification of recombinant LFA-1 αI-Fc fusion protein. The αI domain of LFA-1 was cloned into the pFUSE-hIgG1-Fc2 vector and expressed in Chinese hamster ovary (CHO) K1 cells. The fusion protein was purified from culture supernatants using Protein A affinity chromatography. (i) SDS-PAGE analysis

*Figure 1 continued on next page*

*Figure 1 continued*

of the purified protein revealed a prominent Coomassie-stained band at ~45 kDa, corresponding to the expected molecular weight of the LFA-1 αI-Fc fusion protein. (ii) Western blot analysis using anti-human IgG antibody confirmed the identity of the fusion protein. (**C**) Binding of LFA-1 αI-Fc (250 nM) fusion protein to *P. falciparum*-infected red blood cells (iRBCs). Flow cytometry analysis using PE-Texas Red-conjugated anti-human IgG antibody demonstrated specific binding of the LFA-1 αI-Fc protein to ring, trophozoite, and schizont stages of iRBCs, indicating interaction across all major asexual blood stages.

The online version of this article includes the following source data and figure supplement(s) for figure 1:

**Source data 1.** PDF file containing original uncropped Coomassie Blue-stained SDS-PAGE gel corresponding to *Figure 1Bi* and original uncropped western blot image corresponding to *Figure 1Bii*, with relevant bands and experimental treatments indicated.

**Source data 2.** Original uncropped Coomassie Blue-stained SDS-PAGE gel corresponding to *Figure 1Bi* and original uncropped western blot image corresponding to *Figure 1Bii*.

**Figure supplement 1.** Sequence conservation of the LFA-1 αI domain and validation of its specific binding to *P. falciparum*- infected erythrocytes.

the specificity anti-*Pf*GBP-130 antibodies, as well as of the observed interaction between *Pf*GBP-130 and recombinant LFA-1 αI-Fc protein, thus ruling out non-specific binding artifacts.

## Biophysical assessment of the interaction between *Pf*GBP-130-N and LFA-1 αI-Fc fragments

To confirm the interaction between LFA-1 and PfGBP-130, real-time binding studies were performed using recombinant PfGBP-130N and the LFA-1 αI domain by bio-layer interferometry (BLI) on the Sartorius Octet K2 system. Briefly, the LFA-1 αI domain was immobilized on an AR2G sensor (Octet Amine Reactive Second-Generation biosensor) at 100 ng, and increasing concentrations of *Pf*GBP-130N were applied as analyte. *Pf*GBP-130N displayed a rapid association phase with immobilized LFA-1 αI domain, followed by a clear dissociation phase, consistent with direct binding. To ensure robustness and reproducibility, the BLI experiments were performed in multiple independent replicates (n≥3) using independently purified protein batches. The averaged sensorgrams, together with the standard deviation of the calculated equilibrium dissociation constant, yielded a $K_D$ of $(1.7\pm0.22)\times10^{-8}$ M (*Figure 2Ci*), indicating a strong and specific interaction between LFA-1 αI domain and *Pf*GBP-130N.

Since LFA-1 αI domain displayed a high binding affinity for *Pf*GBP-130 N, we next performed protein-protein docking studies to examine the molecular-level interactions between these two proteins. The *Pf*GBP-130 model was docked against LFA-1 αI domain using the Cluspro2.0 protein-protein docking server, and the binding site for the generated complex was analyzed with PyMOL. The best resulting structure was submitted to PDBSum to identify the amino acid residues involved in the interactions. Examination of the interaction surface showed that LFA1 binds within a curved pocket of *Pf*GBP-130 formed by an anti-parallel helix structure, and the interacting residues for LFA1 were mainly localized in the N-terminal domain. A total of 13 hydrogen bonds and 5 salt bridges were observed for the *Pf*GBP-130/LFA1 complex. The representative hydrogen bonds are shown in *Figure 2Cii*.

Furthermore, the binding energy of the docked LFA1/*Pf*GBP complex was evaluated utilizing several computational tools, including PRODIGY, PPCheck, AREA-AFFINITY, and the HawkDock server, each of which considers different parameters to calculate the energy. As summarized in *Supplementary file 1B*, the calculated energy values suggest stable complex formation.

To assess the strength of the interactions, the buried surface area of the complex was calculated using the PDBePISA web server, yielding a value of 1528.7 Å² per molecule, which is in accordance with the buried area suggested for known protein-protein complexes. Next, we evaluated the stability of the docked complex through molecular dynamics (MD) simulations in an aqueous environment over 200 nanoseconds (ns). The root mean square deviation (RMSD) trajectory plot indicated that after a minor fluctuation during the first 30 ns, the complex remained stable for the rest of the simulation (*Figure 2Ciii*). In conclusion, both real-time BLI and in silico studies suggested a stable interaction between LFA-1 and *Pf*GBP-130.

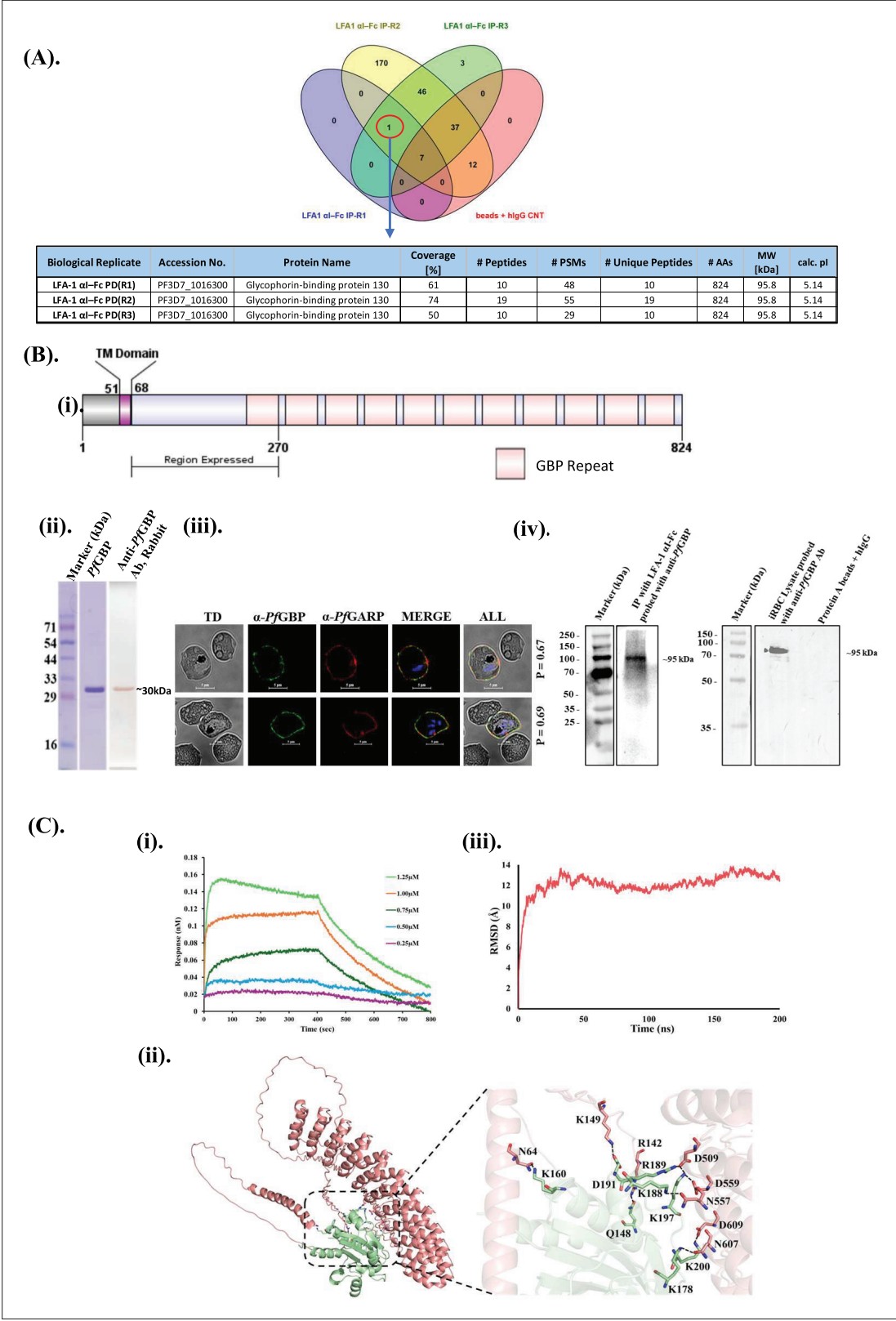

**Figure 2.** *Pf*GBP-130 on the surface of *P. falciparum*-infected erythrocytes binds the LFA-1 αI domain. (**A**) Identification of *Pf*GBP-130 as an interacting partner of LFA-1 αI-Fc. LC-MS/MS analysis of immunoprecipitates from *P. falciparum*-infected erythrocyte lysates pulled down with LFA-1 αI-Fc fusion protein revealed *Pf*GBP-130 (PF3D7_1016300) as a major interacting protein. The table summarizes proteins specifically enriched in the LFA-1 αI-Fc pull-down in all three biological replicates, showing high peptide coverage and spectral abundance, indicating a specific and robust interaction. (**B**)

*Figure 2 continued on next page*

*Figure 2 continued*

Characterization and localization of *Pf*GBP-130. (**i**) Schematic representation of the domain organization of *Pf*GBP-130 and the N-terminal fragment (amino acids 69–270) that was expressed in *Escherichia coli* (termed *Pf*GBP-130-N). (ii) SDS-PAGE and western blot analysis of purified *Pf*GBP-130-N using anti-rabbit *Pf*GBP-130 antibodies. A prominent band at ~30 kDa corresponds to the expected molecular weight of the recombinant fragment. (iii) Immunofluorescence assay (IFA) demonstrating surface localization of *Pf*GBP-130 on trophozoite-stage iRBCs using anti-*Pf*GBP-130 antibodies. *Pf*GBP-130 (green) partially co-localizes with *Pf*GARP (red), a well-established iRBC surface protein with an extracellular domain. Nuclei were stained with DAPI (blue), confirming surface expression. (iv) Western blot analysis of iRBC lysate and of immunoprecipitate of LFA1 αI-Fc-bound iRBC (A) using anti-rabbit GBP-130 antibody. Lane 1 shows the presence of *Pf*GBP-130 in the immunoprecipitate. The ~130 kDa band is notably absent in the control IP eluate where no LFA-1 αI-Fc was bound to iRBC, demonstrating the specificity of the LFA-1 αI-Fc and *Pf*GBP-130 interaction. Lane 2 shows the detection of native *Pf*GBP-130 as an ~110 kDa protein in trophozoite-stage *P. falciparum* lysate, consistent with its predicted molecular weight. (**C**) Biophysical and computational validation of *Pf*GBP-130 and LFA-1 αI interaction. (i) Bio-layer interferometry (BLI) analysis of real-time binding between *Pf*GBP-130-N and LFA-1 αI domain. Averaged sensorgrams across independent experiments (n≥3) demonstrate concentration-dependent association and dissociation kinetics. The calculated equilibrium dissociation constant ($K_D$) was $(1.7\pm0.22)\times10^{-8}$ M, indicating high-affinity binding. (ii) In silico docking analysis showing the energy-minimized complex of *Pf*GBP-130 (salmon) and LFA-1 αI domain (green) generated using ClusPro 2.0. Representative hydrogen bonds and interacting residues are shown as sticks. Visualizations were prepared using PyMOL. (iii) Molecular dynamics (MD) simulation of the *Pf*GBP-130/LFA-1 αI complex. The graph depicts the root mean square deviation (RMSD) over time, confirming structural stability of the protein-protein complex.

The online version of this article includes the following source data and figure supplement(s) for figure 2:

**Source data 1.** PDF file containing original uncropped Coomassie Blue-stained SDS-PAGE gel corresponding to *Figure 2Bii*, and original uncropped western blot images corresponding to *Figure 2Bii* and *Figure 2Biv*, with relevant bands and experimental treatments indicated.

**Source data 2.** Original uncropped Coomassie Blue-stained SDS-PAGE gel corresponding to *Figure 2Bii*, and original uncropped western blot images corresponding to *Figure 2Bii* and *Figure 2Biv*.

**Figure supplement 1.** Cloning and molecular validation of LFA αI-Fc and *Pf*GBP-130 constructs.

**Figure supplement 1—source data 1.** PDF file containing original uncropped agarose gel electrophoresis images corresponding to *Figure 2—figure supplement 1B and C*, with relevant bands and experimental treatments indicated.

**Figure supplement 1—source data 2.** Original uncropped agarose gel electrophoresis images corresponding to *Figure 2—figure supplement 1B and C*.

## *Pf*GBP-130 ectodomain shows a specific LFA-1-dependent binding to human THP-1 monocytic and primary NK cells

*Pf*GBP-130 possesses an N-terminal cytosolic domain, a transmembrane domain, and an extracellular domain characterized by repeats (*Figure 3A*). Since we identified *Pf*GBP-130 as a putative ligand for LFA-1 on NK cells, we next investigated the direct interaction of *Pf*GBP-130 with immune cells, specifically THP-1 monocytes and NK cells. Briefly, a recombinant protein construct comprising the extracellular domain of *Pf*GBP-130, encompassing the putative LFA-1 binding sites and fused to the Fc region of hIgG1 (*Pf*GBP-Fc), was generated (*Figure 3A*). The *Pf*GBP-Fc fusion protein was engineered to ensure stability and solubility for binding assays, while the hIgG1 Fc tag facilitated detection via secondary anti-human IgG antibodies.

The *Pf*GBP-Fc construct was transiently expressed in CHO K1 cells cultured in serum-free medium to minimize potential interference from serum components. The secreted fusion protein was subsequently purified from the culture supernatant using Protein A affinity chromatography (*Figure 3— figure supplement 1A and B*). The interaction between *Pf*GBP-Fc and THP-1 was investigated, focusing on the role of LFA-1 as a potential receptor using siRNAs corresponding to CD11a subunit of LFA-1 mRNA. Next, the cultured THP-1 monocytic cells with and without siRNA treatment were fixed and incubated with recombinant *Pf*GBP-Fc or control hIgG in FACS staining buffer for 2 hr. After washing, cells were stained with fluorescent anti-human PE-Texas Red antibodies (Invitrogen) for binding analysis via flow cytometry to detect the binding of *Pf*GBP-Fc. Flow cytometric analysis demonstrated a significant increase (threefold) in binding of *Pf*GBP-Fc compared to an hIgG isotype control to THP-1 cells, indicating a specific interaction of *Pf*GBP to THP-1 cells (*Figure 3Bi*). To know whether the *Pf*GBP-Fc binding to THP-1 cells is LFA-1 dependent, we next performed SmartPool Accel siRNA-mediated gene silencing of CD11a subunit of LFA-1 in THP-1 cells (*Figure 3—figure supplement 1D*). As shown in *Figure 3Bii*, western blot analysis confirmed efficient depletion of the CD11a subunit of LFA-1 in THP-1 cells. Subsequent flow cytometric analysis revealed a substantial reduction in *Pf*GBP-130-Fc binding to LFA-1-deficient THP-1 cells compared to untreated controls

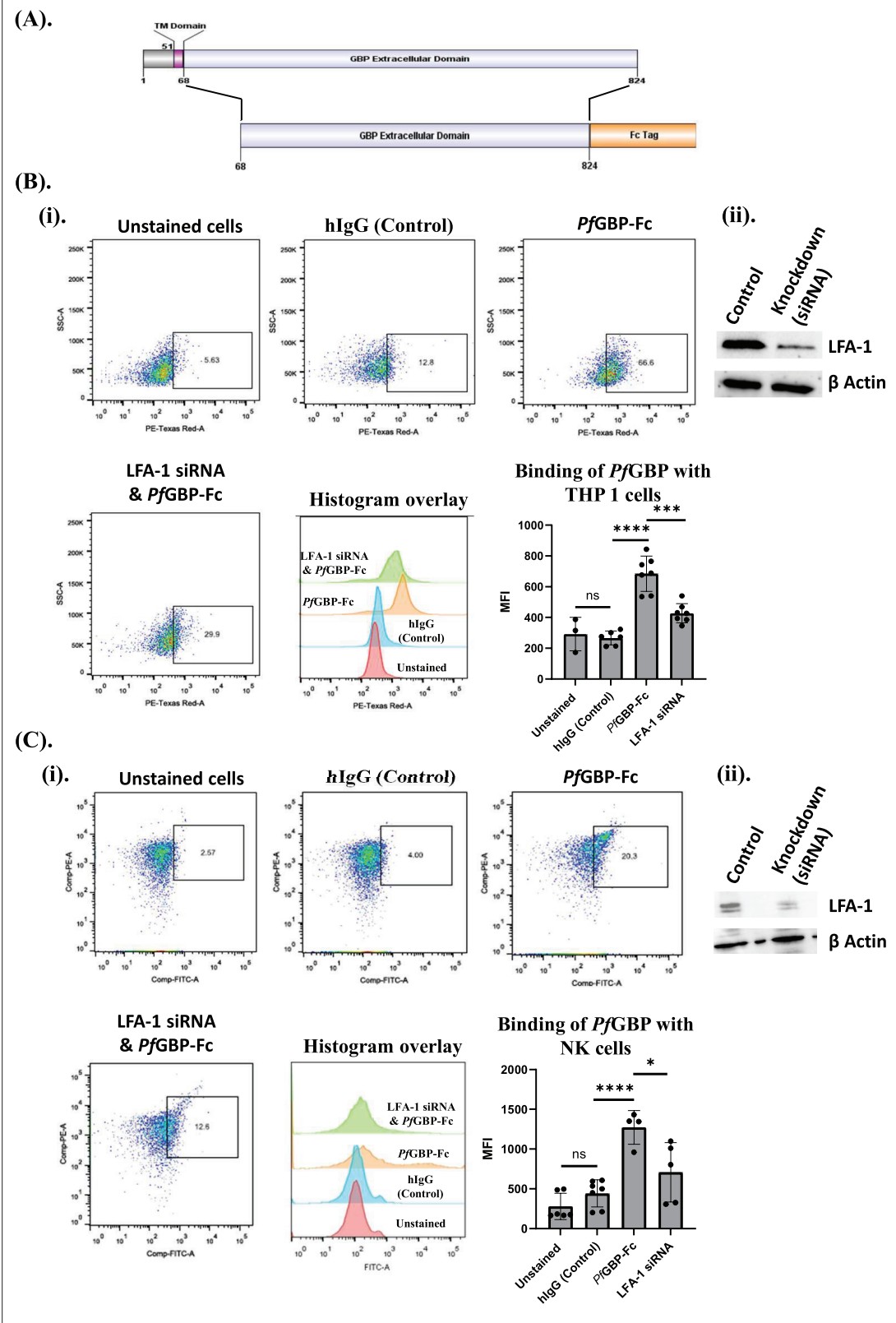

**Figure 3.** Specificity of interaction between LFA-1 on primary NK cells and THP-1 cells with *Pf*GBP-130. Primary NK and THP-1 cells treated with LFA-1 siRNAs showed reduced binding to recombinant extracellular domain of *Pf*GBP-130 expressed in Chinese hamster ovary (CHO) K1 cells. (**A**) Schematic representation of *Pf*GBP-130 and its extracellular domain cloned in pFUSE-hIgG1-Fc2 vector for the expression in CHO K1 cells. The construct comprises the extracellular domain of *Pf*GBP-130 (including putative LFA-1 binding sites) fused to the Fc region of human IgG1. The Fc fusion provides

*Figure 3 continued on next page*

*Figure 3 continued*

stability, solubility, and facilitates detection. (**B**) Interaction of *Pf*GBP-130 ECD-Fc with THP-1 cells. (i) Flow cytometric analysis shows strong binding of *Pf*GBP-130 ECD-Fc to THP-1 cells, compared to an hIgG1 isotype control, indicating specific interaction. LFA-1 knockdown in THP-1 cells via siRNA treatment significantly reduced PfGBP-130 ECD-Fc binding, thus confirming the specificity of interaction between *Pf*GBP-130 ECD-Fc and LFA-1 on THP cells. (ii) Western blot analysis using anti-CD11a antibody confirmed the siRNA-mediated knockdown of CD11a subunit of LFA-1 protein on THP-1 cells. (**C**) Interaction of *Pf*GBP-130 ECD-Fc with primary human NK cells. (i) Flow cytometry revealed a marked increase in *Pf*GBP-130 ECD-Fc binding to NK cells over the isotype control. siRNA-mediated knockdown of LFA-1 in NK cells led to a notable reduction in *Pf*GBP-130 ECD-Fc binding, confirming that LFA-1 is essential for this interaction. (ii) Western blot analysis using anti-CD11a antibody confirmed the siRNA-mediated knockdown of CD11a subunit of LFA-1 protein in NK cells, confirming LFA-1 as a critical receptor for *Pf*GBP-130 ECD-Fc binding to both NK and THP-1 cells. * Denotes $p<0.05$, ** denotes $p<0.01$, and *** denotes $p<0.001$. Representative flow plots depict the percentage of cells within a predefined positive gate, whereas the accompanying summary graph quantifies fluorescence intensity across the analyzed population. These two metrics report distinct properties of the distribution and are therefore not expected to be numerically identical.

The online version of this article includes the following source data and figure supplement(s) for figure 3:

**Source data 1.** PDF file containing original uncropped western blot images corresponding to *Figure 3Bii and Cii*, with relevant bands and experimental treatments indicated.

**Source data 2.** Original uncropped western blot images corresponding to *Figure 3Bii and Cii*.

**Figure supplement 1.** Expression and validation of recombinant *Pf*GBP-130-Fc protein and siRNA-mediated inhibition of LFA-1 (CD11a) expression in immune cells.

**Figure supplement 1—source data 1.** PDF file containing original uncropped Coomassie Blue-stained SDS-PAGE gel corresponding to *Figure 3— figure supplement 1A* and original uncropped western blot image corresponding to *Figure 3—figure supplement 1B*, with relevant bands and experimental treatments indicated.

**Figure supplement 1—source data 2.** Original uncropped Coomassie Blue-stained SDS-PAGE gel corresponding to *Figure 3—figure supplement 1A* and original uncropped western blot image corresponding to *Figure 3—figure supplement 1B*.

**Figure supplement 2.** *Pf*GBP-130-Fc binds specifically to LFA-1 expressing cells but not to negative control cell lines.

(*Figure 3Bi*). This reduction thus definitively established LFA-1 as a crucial mediator of *Pf*GBP-Fc binding to THP-1 monocytic cells.

We further extended these findings to other immune cell populations, such as primary NK cells. Consistent with the results obtained in THP-1 cells, incubation of NK cells with *Pf*GBP-Fc resulted in a significant increase in binding intensity relative to the hIgG1 isotype control (*Figure 3Ci*). Smart-Pool Accel siRNA-mediated downregulation of CD11a subunit of LFA-1 in NK cells (*Figure 3Cii* and *Figure 3—figure supplement 1C*) reduced *Pf*GBP-130-Fc binding to NK cells (*Figure 3Cii*), thereby confirming the role of LFA-1 as a critical receptor for *Pf*GBP-Fc binding to NK cell.

LFA-1 is predominantly expressed on immune cells; to further confirm ligand specificity of LFA-1, we evaluated the binding of *Pf*GBP-130-Fc to multiple non-immune cell types, including HEK293T cells, HepG2 cells, and adipose-derived stem cells (ADSCs), which exhibit no or only very low basal expression of LFA-1. THP-1 cells, which express LFA-1, were included as a positive control. In contrast to THP-1 cells, all three non-immune cell types displayed minimal to negligible binding of *Pf*GBP-130-Fc (*Figure 3—figure supplement 2A–D*). Notably, HepG2 cells and stem cells showed only background-level binding comparable to the hIgG isotype control. These findings support that *Pf*GBP-130-Fc binding is LFA-1 dependent and restricted to immune cells expressing LFA-1, reinforcing the specificity of the interaction.

## Engineered chimeric GBP expressing CHO cell line activates NK cell through LFA-1 binding

To elucidate the mechanism by which *P. falciparum* iRBCs activate NK cells through LFA-1 engagement, we hypothesized that *Pf*GBP-130 serves as a cognate ligand for LFA-1 on the iRBC surface. To test this hypothesis, a CHO cell line was engineered for the stable expression of *Pf*GBP-130 extracellular domain on its surface by leveraging lentiviral transduction. The engineered CHO K1 expressed the extracellular C-terminal domain of *Pf*GBP-130 fused to the transmembrane domain of transferrin receptor (TfR-TM) for the surface presentation (*Figure 4—figure supplement 1A*). The surface expression was verified using anti-*Pf*GBP-130 antibodies by immunofluorescence (*Figure 4Ai*). Flow cytometry analysis demonstrated robust binding of recombinant LFA-1 αI-Fc to *Pf*GBP-130 expressing CHO cells, while no significant binding was observed with mock-transduced CHO cells, confirming the specificity of the interaction (*Figure 4Aii*).

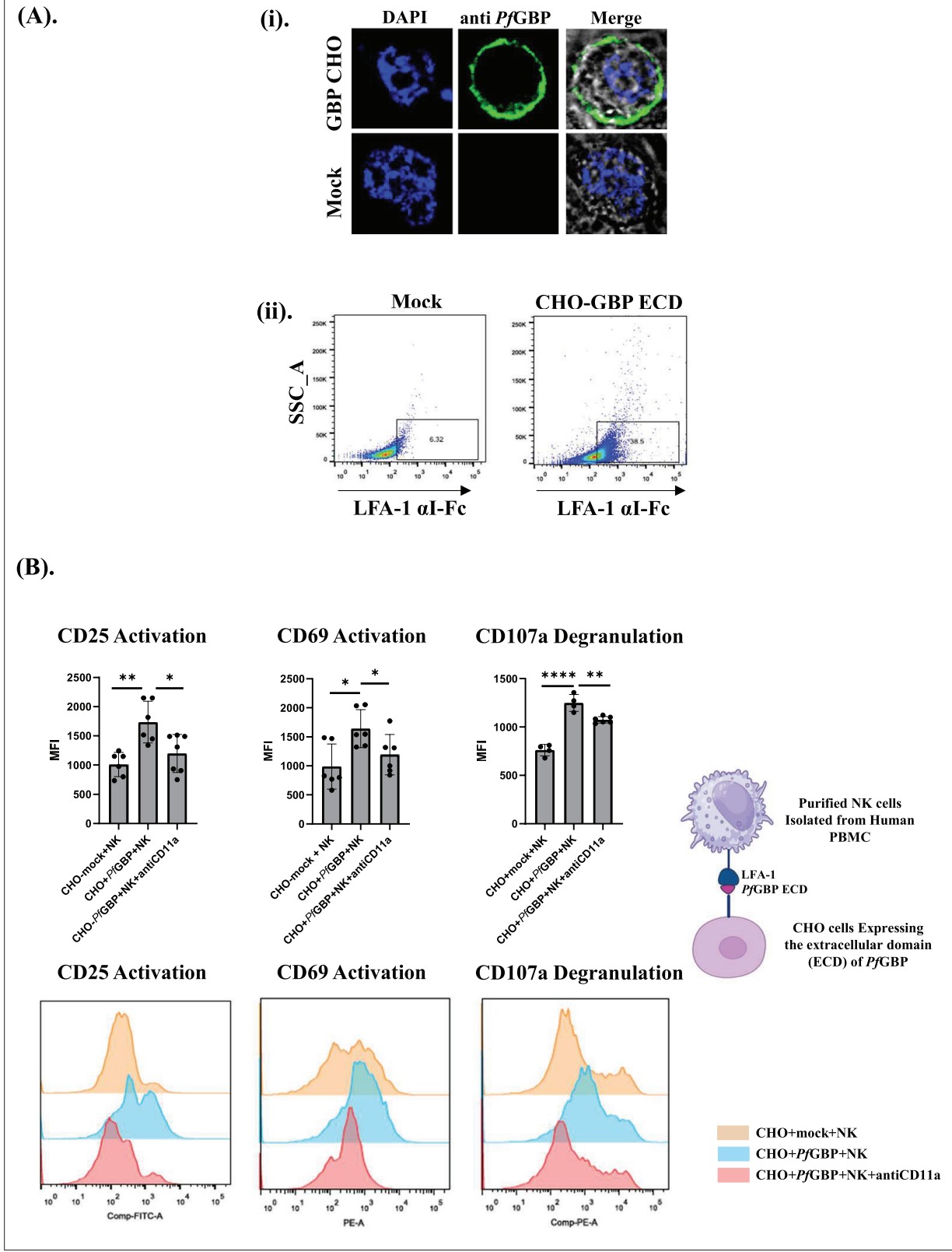

**Figure 4.** Natural killer (NK) cells activated in the presence of *Pf*GBP-130. (**A**) (i) Expression of *Pf*GBP-130 ECD fused with transferrin membrane domain on the membrane of Chinese hamster ovary (CHO) K1 cells by infecting the lentiviral vector; pMSCV-puro and its immunofluorescence analysis using anti-rabbit *Pf*GBP antibody. (ii) CHO K1 cells expressing *Pf*GBP-130 ECD bind LFA-1 αI-Fc. Binding of purified LFA-1 αI-Fc to *Pf*GBP-130 ECD expressing CHO cells was assessed by FACS using a PE-Texas Red anti-human IgG antibody. (**B**) NK cell activation in the presence of CHO K1 cells expressing

*Figure 4 continued on next page*

*Figure 4 continued*

*Pf*GBP ECD. Human NK cells were purified (>95%) from fresh peripheral blood mononuclear cell (PBMC) and co-cultured with CHO K1 cells expressing *Pf*GBP ECD in a 2:1 ratio (20,000 CHO K1 cells:10,000 NK cells), and these cells were stimulated with (Poly I:C/Lipofectamine 2000) for 24 hr. NK cells were separated from adherent CHO K1 cells, and NK cell activation was assessed by assaying the expression of activation markers (CD69, CD25) and a degranulation marker, CD107a. NK cells co-cultured with CHO K1 cells expressing *Pf*GBP-ECD protein showed significant increase in the expression of CD25 and CD69, as well as CD107a in comparison to the NK cells co-cultured with mock CHO cells. Addition of anti-CD11a (HI111 clone) antibodies reduced the expression of both activation and degranulation markers. * Denotes p<0.05, ** denotes p<0.01, and *** denotes p<0.001.

The online version of this article includes the following figure supplement(s) for figure 4:

**Figure supplement 1.** Schematics of the *Pf*GBP-130-ECD lentiviral expression vector and characterization of primary NK cell purity.

To further investigate the role of *Pf*GBP-130 in NK cell activation and degranulation via direct engagement with its cognate receptor LFA-1, we co-cultured CHO cells stably expressing the extracellular domain of *Pf*GBP-130 (CHO-*Pf*GBP-ECD) with primary human NK cells at a 2:1 ratio (CHO-*Pf*GBP-ECD:NK). Given that LFA-1 affinity is dynamically regulated through inside-out signaling, co-cultures were stimulated with Poly I:C/Lipofectamine 2000 (synthetic double-stranded RNA analog) to induce a high-affinity conformational state of LFA-1. Subsequently, NK cell activation was evaluated by flow cytometric detection of early activation markers CD69 and CD25. Compared to mock CHO cells, *Pf*GBP-130 expressing CHO cells elicited a significant ~50% increase in CD69 and ~40% increase in CD25 expression, indicating enhanced NK cell activation by *Pf*GBP-130. This activation was specific as addition of anti-CD11a antibodies in co-cultures reduced this activation significantly (*Figure 4B*).

Furthermore, we examined the impact of *Pf*GBP-130 expression on primary NK cells degranulation, an indirect indicator of cytotoxicity. For this surface expression of CD107a, a marker of lysosomal degranulation was measured using flow cytometry. Co-culture of NK cell with CHO-*Pf*GBP-ECD resulted in a substantial increase in CD107a on primary NK cells as compared to mock CHO control. This increase was significantly reduced in the presence of anti-CD11a antibodies, indicating that the enhanced degranulation was specifically mediated through LFA-1 engagement with *Pf*GBP-130 on the CHO cell surface (*Figure 4B*). Together, these results indicate that *Pf*GBP-130 protein promotes LFA-1-mediated activation and cytotoxic degranulation of NK cells.

## Human NK cells eliminate iRBCs in a co-culture study, and this elimination is dependent on GBP-130 expression on iRBCs

To gain direct evidence of NK cell-mediated killing of iRBCs via *Pf*GBP-130-dependent interaction, we performed co-culture experiments using purified human NK cells and iRBCs as described earlier by *Chen et al., 2014*, in the presence or absence of anti-*Pf*GBP-130 antibodies. An isotype control antibody served as a negative control for non-specific antibody effects. Briefly, isolated human NK cell was treated with Fc receptor blocker using anti-CD16 (3G8 clone) prior to addition of iRBC (0.5% parasitemia) in a 10:1 ratio (NK:iRBC). The blocking of CD16 with anti-CD16 (clone 3G8) inhibits Fc engagement with CD16 receptor and thereby abrogates any potential ADCC contribution during the NK-iRBC co-culture in the presence of anti GBP neutralizing antibody (*Mandelboim et al., 1999*; *Yeap et al., 2016*). The co-cultures were maintained for 48 hr and 96 hr, a time frame sufficient for NK cell activation and parasitemia control, respectively.

In a parallel set of experiments, we sought to specifically examine the effect of blocking *Pf*GBP-130 on iRBC with anti-*Pf*GBP neutralizing antibody on NK cell activation and parasitemia control. The purity of isolated NK cells was accessed using anti-CD3 and anti-CD56 (*Figure 4—figure supplement 1B*). Isolated NK cells were co-cultured with iRBCs for an extended period of 96 hr to allow for a more comprehensive assessment of parasite growth, which was assayed by Hoechst staining, and NK cell numbers were assessed by anti-CD56 NK antibody staining in a flow cytometry analysis. The iRBCs were positive for Hoechst, but negative for CD56, while NK cells were positive for both. The co-culture of NK cells with iRBC either alone or in the presence of rabbit IgG (rIgG) isotype control antibody significantly reduced parasitemia compared to iRBCs cultured in the absence of NK cells. In contrast, the addition of anti-*Pf*GBP antibody to NK cell-iRBC co-cultures resulted in a marked and statistically significant increase in parasitemia, indicating that blockade of *Pf*GBP-130 impairs NK cell-mediated parasite clearance (*Figure 5A*). Conversely, parasitemia levels in the co-cultures supplemented with the isotype control antibody were observed to be comparable to those in the NK cell culture alone, indicating that the observed reduction in parasitemia control was specifically attributable to the

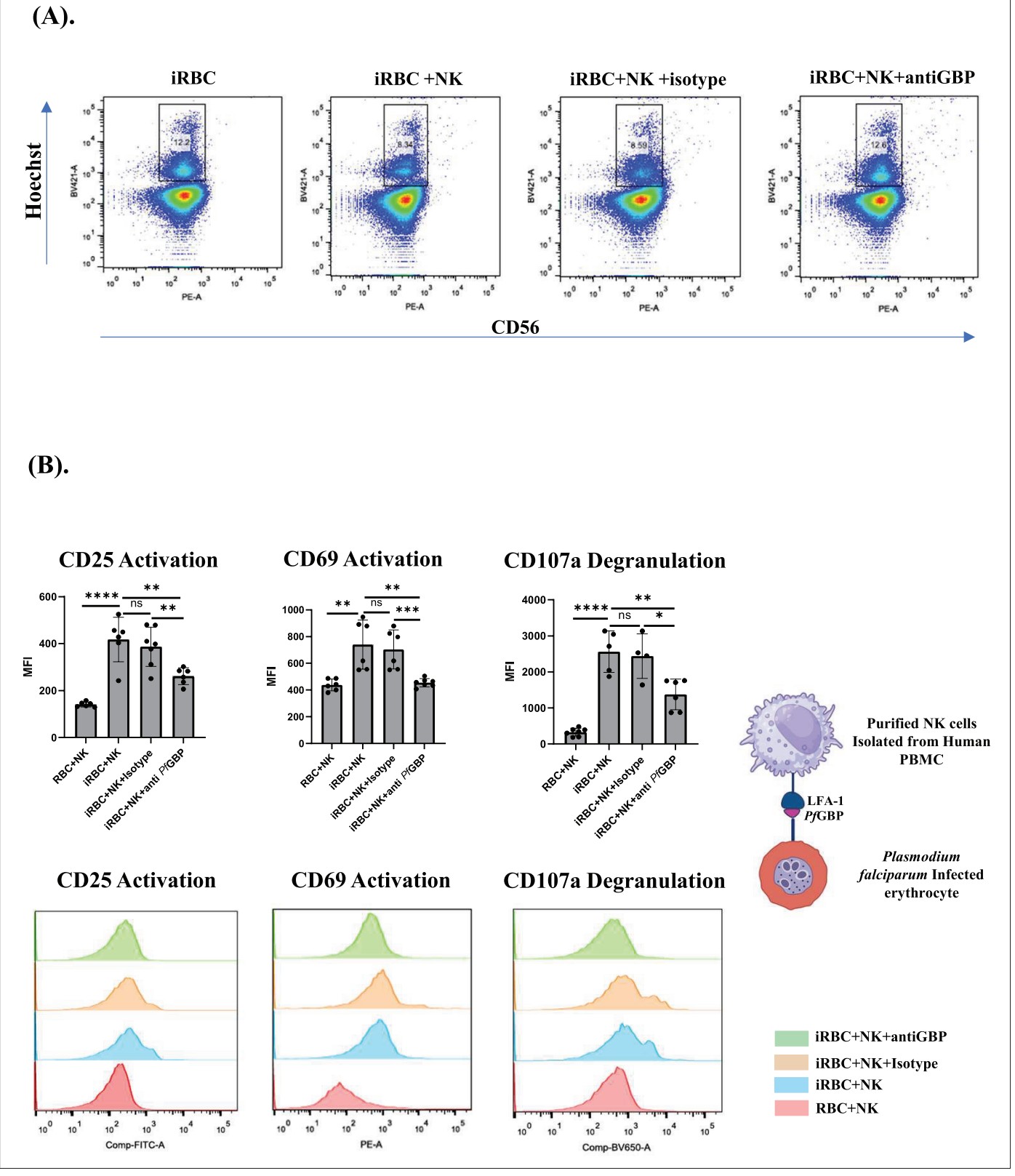

**Figure 5.** Natural killer (NK) cells activated in the presence of infected red blood cells (iRBCs) control parasite infection. (**A**) Activated human NK cells eliminate iRBCs in vitro. Human NK cells when co-cultured with iRBCs reduce parasitemia significantly after 96 hr, and in the presence of anti-*Pf*GBP-130 antibodies, this reduction in parasitemia was blocked. Presence of anti-GBP-130 antibodies resulted in parasitemia similar to the control when NK cells were incubated with iRBCs alone. (**B**) NK cell activation in the presence of iRBCs. Human NK cells were purified (>95%) from fresh peripheral

*Figure 5 continued on next page*

*Figure 5 continued*

blood mononuclear cells (PBMCs) and co-cultured with synchronized schizont-stage iRBCs at a parasitemia of 0.5% in a 10:1 ratio (NK:iRBC) for 48 hr. Quantification of activation and degranulation markers was performed after 48 hr. NK cells co-cultured with iRBCs showed significant increase in the expression of CD25 and CD69, the two activation markers, as well as for the expression of CD107a, a degranulation marker in comparison to the NK cells co-cultured with RBCs alone. Addition of anti-rabbit PfGBP-130 antibodies reduced the expression of both activation and degranulation markers in these NK cells in comparison to rabbit IgG isotype control. * Denotes p<0.05, ** denotes p<0.01, and *** denotes p<0.001.

blockade of *Pf*GBP-130 and not due to non-specific antibody interactions (*Figure 5A*). This finding confirms the specificity of the anti-*Pf*GBP antibody and reinforces the critical role of *Pf*GBP in NK cell-mediated parasite clearance.

We further studied the levels of activation and degranulation markers in NK cells in the absence and presence of anti-GBP-130 antibody (*Figure 5B*). Briefly, NK cells and iRBC were co-cultured in the presence of anti-GBP-130 antibody or rIgG isotype control antibody. Incubation of human NK cells with iRBC for 48 hr showed a significant percent change in MFI of activation markers, CD69 and CD25, as well as degranulation marker, CD107a. This effect was reduced in the presence of anti-*Pf*GBP antibody, which signified that *Pf*GBP-130 engagement is necessary for the activation and degranulation of NK cells (*Figure 5B*). These results were specific as levels of activation and degranulation markers were same in NK cells incubated with iRBC in the presence or absence of rIgG control. Taken together, the reduced NK cell activation and increase in parasitemia following *Pf*GBP-130 blockade strongly support the conclusion that direct interaction between LFA-1 on NK cells and GBP-130 on iRBCs is a critical determinant of contact-dependent NK cell-mediated immune responses. These findings further establish *Pf*GBP-130 as a key ligand in the LFA-1-dependent activation of NK cells during *P. falciparum* infection.

## Discussion

NK cells have been well described for their ability to eliminate viral-infected cells, tumors, and several pathogens, including *P. falciparum* (*Burrack et al., 2019*). A fine-tuned panel of activating and inhibitory receptors regulates the activation of NK cells. Among these, LFA-1, an integrin family member, serves not only as an adhesion molecule but also as an important activating receptor on NK cells (*Urlaub et al., 2017*). Beyond mediating adhesion, LFA-1 signaling is essential for the polarization of lytic granules toward target cells, a prerequisite for efficient NK cell cytotoxicity (*Bryceson et al., 2005*; *Kabanova et al., 2018*).

There is mounting evidence to show that NK cells contribute to immune responses in the clearance of parasites, removal of infected hepatocytes, and iRBCs through cytotoxicity and ADCC (*Burrack et al., 2019*). Using an immunodeficient humanized RAG-IL2Rγc-deficient (RICH) mice, a study by *Chen et al., 2014* showed that NK cells, rather than macrophages, are responsible for regulating the parasite growth in vivo by directly interacting with iRBCs. This study also implicated LFA-1 as a key receptor involved in NK cell-mediated killing of iRBCs. LFA-1 is a heterodimer composed of CD11a/CD18, also called α1/β2. The LFA-1α subunit, αL, has two prominent structural features, I domain of 200 aa residues and three EF hand-like domains, which are crucial for ligand binding. Two other integrins expressed on leukocytes, Mac-1 (CD11b/CD18) and p150.95 (CD11c/CD18), share the same CD18 or β2 integrin subunit and have homologs α subunits (*Kolanus et al., 1996*). The I domain of these proteins is homologous to motifs in von Willebrand factor, cartilage matrix protein, collagen type VI, complement factor C2, and factor B (*Huang and Springer, 1995*). Structure-function studies on LFA-1 and Mac-1 have implicated I domains in ligand binding (*McDowall et al., 1998*). Here, we set out experiments to know whether LFA-1 αI, a 200 aa domain of LFA-1 on NK cells, alone recognizes the iRBCs. To do so, LFA-1α1 domain was expressed as a fusion protein with Fc domain of hIgG and was allowed to bind *P. falciparum*-infected erythrocytes. LFA-1α1-Fc fusion protein bound significantly with iRBCs in comparison with hIgG control protein.

Having established LFA-1α1 as a key binding domain to iRBCs, we next carried out studies to identify the parasite-derived ligand on iRBC surface that interacts with LFA-1. Up to now, LFA-1 has been shown to bind human and mouse ICAM-1 (*Huang and Springer, 1995*). A unique property of LFA-1 is that its binding affinity can be dynamically modulated by inside-out (intracellular signals triggered by antigens and chemokines) and outside-in (ligand binding) signaling pathways (*Gérard et al., 2021*;

*Kondo et al., 2022*). Conformation changes and membrane clustering of LFA-1 have been shown to significantly influence its binding affinity (*Shi and Shao, 2023*; *Urlaub et al., 2017*). To fish out the parasite ligand on the surface of iRBCs for LFA-1, here, we cross-linked LFA-1α1-Fc fusion protein with iRBCs, followed by immunoprecipitation and LC-MS/MS analysis. *Pf*GBP-130 protein was identified as a prominent LFA-1 interacting protein on the iRBC surface. As the name suggests, *Pf*GBP-130 is a glycophorin binding protein consisting of 11 highly conserved 50 amino acid repeats and a charged N-terminal region of 225 amino acids (*Kochan et al., 1986*). A number of studies have shown that *Pf*GBP-130 is exported out of iRBC surface and is involved in erythrocyte invasion (*Soni et al., 2016*). We further confirmed the iRBCs surface expression of *Pf*GBP-130 protein using anti-*Pf*GBP antibodies, thus supporting the observation that it probably is involved in recognition of iRBCs by NK cells. Western blot analysis of LFA-1 αI -Fc-bound iRBCs lysate using anti-*Pf*GBP antibodies further strengthened the hypothesis that *Pf*GBP-130 on iRBCs surface is the ligand for LFA-1 that helps NK cells to recognize the iRBCs. To evaluate the specificity of LFA-1 interaction with *Pf*GBP-130 on iRBC surface, we overexpressed external domain of *Pf*GBP-130 as a Fc fusion protein and evaluated its binding to THP-1 monocytes and primary NK cells. Recombinant *Pf*GBP-Fc protein efficiently bound to the surface of both cell types. Importantly, siRNA-mediated knockdown of CD11a subunit LFA-1 significantly reduced this binding, confirming that *Pf*GBP-130 directly interacts with LFA-1 on immune cells, thereby advocating GBP-130 as a ligand for LFA-1.

NK cells are predominantly cytolytic lymphocytes, and their primary effector function relies on the degranulation of cytotoxic molecules. LFA-1 not only mediates tight adhesion to target cells but also plays a crucial role in early NK cell activation and degranulation (*Barber et al., 2004*). To assess whether *Pf*GBP-130 triggers NK cell activation through LFA-1 engagement, we analyzed the expression of activation markers CD25 and CD69, and the degranulation marker CD107a, in NK cells co-cultured with CHO cells expressing the extracellular domain of *Pf*GBP-130. Compared to mock controls, co-culture with *Pf*GBP-130 expressing CHO cells significantly upregulated all three markers. Importantly, this activation was abrogated by the addition of anti-CD11a antibodies, confirming that *Pf*GBP-130 mediates NK cell activation and degranulation via LFA-1. CD107a (LAMP-1), a lysosomal membrane protein, serves as a well-established marker of NK cell degranulation (*Urlaub et al., 2017*; *Ye et al., 2018*).

Besides mediating the tight adhesion of NK cells to various target cells, LFA-1-mediated signaling plays a pivotal role in the polarization of lytic granules toward the immunological synapse, thereby enhancing NK cell-mediated cytotoxicity (*Barber et al., 2004*; *Shi and Shao, 2023*). To know whether such toxicity is also generated as a result of interaction between NK cells and iRBCs, we analyzed the expression of two activation markers, CD25 and CD69, and a degranulation marker, CD107a, in NK cells co-cultured with CHO cells overexpressing *Pf*GBP-130 protein on their surface as described by *Ye et al., 2018*. All these three markers showed significant upregulation in primed NK cells in comparison to naïve NK cells. This upregulation was specific as the presence of anti-LFA-1 antibodies reversed these upregulations. To determine whether the activation of primary NK cells is specifically mediated by direct engagement between LFA-1 on NK cells and *Pf*GBP-130 on iRBCs at the immunological synapse, we performed targeted co-culture assays. We co-cultured purified human NK cells with iRBCs and measured both parasite growth and the levels of activation and degranulation markers in NK cells, in the presence or absence of anti-*Pf*GBP-130 antibodies. The results of co-culture study showed that the presence of iRBCs resulted in activation of NK cells, and this resulted in reduced parasitemia. This activation of NK cells was as a result of *Pf*GBP-130 expression on iRBCs as anti-*Pf*GBP-130 antibodies reduced the activation of NK cells. These results are in line with a recent study that showed upregulation of these markers in NK cells co-cultured with iRBCs from human population (*Arora et al., 2018*; *Ye et al., 2018*).

In conclusion, the results presented here identify *Pf*GBP-130 as a novel ligand for LFA-1, particularly for the LFA-1 αI domain on NK cells. This interaction mediates firm adhesion between human NK cells and *P. falciparum* iRBCs resulting in the activation of NK cells. The ensuing activation subsequently triggers NK cell cytotoxic degranulation, resulting in targeted killing of iRBCs. Given that NK cells constitute the first line of defense against malaria, this molecular mechanism appears to be important in early control of *P. falciparum* infection, particularly by implementing a host-directed therapy.

## Materials and methods

### *Plasmodium* culture of 3D7

*P. falciparum* 3D7 parasites were maintained in O+ve human erythrocytes at 4% hematocrit in RPMI 1640 medium (pH 7.4) supplemented with 50 mg/L hypoxanthine, 5% Albumax II, 2 g/L sodium bicarbonate, and 20 µg/mL gentamycin according to the protocol described by *Trager and Jensen, 1976*. Cultures were maintained at 37°C in a gas mixture of 90% $N_2$, 5% $CO_2$, and 5% $O_2$. Parasite synchronization was performed using 5% (wt/vol) sorbitol.

### Cell lines and primary human NK cell culture

THP-1 cells, CHO K1, and HEK293T cells were cultured in RPMI 1640 (Invitrogen) supplemented with 10% fetal bovine serum (FBS) or in DMEM (for HEK293T) with 10% FBS. Primary human NK cells were isolated from peripheral blood mononuclear cells (PBMCs) by negative selection using the BioLegend NK Cell Isolation Kit, achieving >95% purity. ADSC was cultured in commercial serum-free media MesenCult-XF as per the manufacturer's instructions. The cell lines were obtained from collaborators and were originally sourced from American Type Culture Collection (USA). Authentication of the cell lines was performed at ATCC using STR profiling. In our laboratory, cells were used at low passage and were routinely tested for mycoplasma contamination.

### Fusion protein construct and its expression and purification

The LFA-1 αI domain was PCR-amplified from THP-1 cDNA and cloned into the pFUSE-hIgG Fc2 vector (InvivoGen). The construct was transfected into CHO K1 cells using JetPrime (Polyplus). Post-transfection, cells were maintained in IgG-depleted FBS-containing media and later switched to 10% Nuserum medium. After 72 hr, the culture supernatant was collected, filtered, and the secreted fusion protein purified using Protein A agarose affinity chromatography (G-Biosciences). Proteins were concentrated using 3 kDa Centricon filters (Merck) and quantified via BCA assay.

### Immunoprecipitation and mass spectrometry

Schizont-stage *P. falciparum* iRBCs were isolated via Percoll gradient centrifugation and incubated with 100 µg LFA-1 αI-Fc or human IgG control for 2 hr at 37°C. Cross-linking was performed using 5 mM DTSSP (Thermo Fisher) for 30 min. Cells were lysed with IP Native Lysis Buffer and incubated with Protein A magnetic beads (Thermo Fisher) for 2 hr. Beads were washed and proteins eluted with 5 mM DTT. Eluates were processed for LC-MS/MS analysis. Eluates were processed for LC-MS/MS analysis. The eluates were first precipitated using 10% trichloroacetic acid, 5% acetone, and 5 mM DTT. Samples were air-dried and resuspended in 100 mM ammonium bicarbonate (ABC) buffer containing 8 M urea, then diluted to 1 M urea with 100 mM ABC buffer. Followed by reduction with 10 mM DTT for 1 hr at room temperature (RT) and alkylation with 40 mM iodoacetamide (Sigma-Aldrich) for 1 hr at RT in the dark. The alkylated eluates were further digested with trypsin (1:50, wt/wt) at 37°C for 12–16 hr. Peptides were acidified with 0.1% formic acid and analyzed on an Orbitrap Fusion Lumos mass spectrometer coupled to a Nano-LC 1200 (Thermo Fisher Scientific). Data were processed with Proteome Discoverer v2.4 using the SEQUEST and AMANDA algorithms.

### iRBC and immune cell binding assay

*P. falciparum* 3D7 parasites were cultured to 5% parasitemia and synchronized. Ring, trophozoite, and schizont stages were isolated using Percoll gradient. Approximately $10^6$ iRBCs from each stage were incubated with 250 nM purified LFA-1 αI-Fc fusion protein or 250 nM hIgG control for 2 hr. After washing, cells were stained with PE-Texas Red or FITC-conjugated anti-human Fc antibodies (Invitrogen) for 30 min and analyzed by flow cytometry. Similarly, THP-1 cells and primary human NK cells, including siRNA-treated populations, were fixed and incubated with GBP130-Fc (250 nM) or hIgG control (250 nM) in FACS buffer for 2 hr. After washing, cells were stained with PE-Texas Red-conjugated anti-human secondary antibodies and analyzed by flow cytometry.

### Cloning, expression, and purification of *Pf*GBP-130 (PF3D7_1016300)

PF3D7_1016300 (aa 69–270; encompassing one GBP repeat) was amplified from *P. falciparum* genomic DNA/cDNA, cloned into pGEM-T, and confirmed by restriction digestion and sequencing

(*Figure 2—figure supplement 1*). The verified fragment was subcloned into pET28b between the NcoI/XhoI sites and transformed into *E. coli* BL21(DE3). Recombinant expression was induced with 1 mM IPTG for 5 hr at 37°C, and cells were lysed by sonication in Tris buffer (50 mM Tris, pH 8.0, 150 mM NaCl). Soluble and insoluble fractions were separated by centrifugation, and expression was assessed by SDS-PAGE and anti-His immunoblotting. His-tagged *Pf*GBP-130 was purified from lysates using Ni-NTA affinity chromatography, with wash steps in lysis buffer containing 10 mM imidazole and elution using a 50–500 mM imidazole gradient. Polyclonal antisera against purified recombinant fragments were raised in BALB/c mice and New Zealand White rabbits as described previously (*Sachdeva et al., 2006*).

## In vitro *Pf*GBP-130/LFA-1 αI-Fc interaction analysis with BLI, in silico docking, and MD stimulation analysis

The real-time interactions between LFA-1 and its potential binding partner were studied using BLI on the Sartorius Octet K2 system. In this setup, LFA1 was immobilized onto an AR2G sensor (Octate Amine Reactive Second-Generation biosensor) at a concentration of 100 ng, and the unbound protein was washed off using 1× phosphate-buffered saline (PBS) pH 7.4 buffer. To determine the binding affinity, *Pf*GBP was tested at various concentrations. Throughout the experiment, 1× PBS pH 7.4 was used as both the running and the dissociation buffer, and all working dilutions of *Pf*GBP proteins were prepared in the same buffer. For data normalization, a control sensor with immobilized LFA1 was run in parallel, where only the running buffer (1× PBS) flowed over the sensor. The interaction kinetics, including the association and dissociation curves, were monitored for 800 s in a series of increasing concentrations of the *Pf*GBP protein. The data was acquired with Octet BLI Discovery 12.2 software, and the KD value was calculated through the Octet Evaluation software, applying a 1:1 binding model for fitting the curve. The experiment was performed at 25°C.

In silico docking and simulation approaches were employed to investigate the molecular interaction between LFA1 and *Pf*GBP. To achieve this, initially, a homology model for PfGBP was generated using AlphaFold (*Jumper et al., 2021*), and the quality of the generated model was assessed with PROCHECK and PyMOL. While for LFA1, its crystal structure (PDB code: 1LFA) was employed in the interaction studies. Next, rigid body protein-protein docking was carried out using the Cluspro2.0 web server (*Comeau et al., 2004*), a top performer at CAPRI (Critical Assessment of Predicted Interactions) challenges. ClusPro ranked the docked models based on cluster size and energy. The default set of docking parameters was applied, and the top-ranked complexes were selected based on the largest cluster sizes and lowest energy values. The number of cluster members and the model cluster score (i.e. cluster center and lowest energy) are shown in *Supplementary file 1B*. The cluster center weighted score shows the structure in the cluster that has the highest number of neighboring structures, whereas the lowest-energy score represents the structure which has the lowest energy in the cluster.

The models were further analyzed to examine the interaction interface using PyMOL (PyMOL Molecular Graphics System, v2.1 by Schrödinger, LLC). The final docked model for the LFA1/*Pf*GBP complex was chosen based on significant interactions and the model with the heighest buried surface area (BSA). PDBsum software was utilized to analyze the interactions within the complex (*Laskowski et al., 2018*).

To examine the stability, the LFA1/*Pf*GBP complex was subjected to MD simulations using the Desmond module of Schrödinger software. First, the Schrödinger protein preparation tool was used to prepare the protein: by removing the non-bonded water (>3 Å from protein residues), optimizing the H-bonds, and energy minimizing the final structures using OPLS3. Next, these complexes were placed in an orthorhombic water box at a buffer distance of 10 Å, and solvated using the TIP4PEW solvation model while maintaining a NaCl concentration of 0.15 M to ensure a physiological ionic strength. After solvation, the complex was subjected to energy minimization using OPLS4 force field parameters followed by relaxation, and the simulation was performed for 200 ns at NPT conditions with a frame captured at every 0.2 ns. For determining the trajectories, the frames were collected and examined through a simulation interaction diagram, which helped in determining fluctuations (RMSD). The final frame of the simulation was saved in a PDB file format. Further, PDBePISA (Protein Interfaces, Surfaces, and Assemblies [PISA]) server was employed to analyze the intermolecular interfaces (*Krissinel and Henrick, 2007*), including computing BSA (Å) for the complex. PyMol was used

to represent these interactions, while binding energy estimations were performed employing multiple computational tools (*Sukhwal and Sowdhamini, 2015*; *Weng et al., 2019*; *Xue et al., 2016*; *Yang et al., 2023*).

## Chimeric CHO stable cell line generation

To generate CHO K1 cells stably expressing *Pf*GBP-130, the codon-optimized synthetic gene encoding the extracellular domain of *Pf*GBP-130 was cloned into the lentiviral transfer plasmid pMSCV-puro. Lentiviral particles were produced by co-transfecting HEK293T cells with pMSCV-puro-*Pf*GBP-130, psPAX2 (packaging plasmid), and pMD2.G (VSV-G envelope plasmid) using JetPrime transfection reagent (Polyplus), following the manufacturer's instructions. After 72 hr, viral supernatants were harvested and concentrated using Lentivirus Concentrator Solution (Takara). CHO K1 (ATCC, USA) cells were transduced with the concentrated lentivirus for 48 hr to facilitate viral entry and integration of *Pf*GBP-130 gene, and selected with puromycin (5 µg/mL) for 7 days to enrich for stably transduced cells. Following antibiotic selection, cells were expanded in fresh culture medium, and *Pf*GBP-130 surface expression was confirmed by fluorescence microscopy.

## NK cell activation assay

*Pf*GBP-130-expressing CHO stable cells were trypsinized and counted before being co-incubated with purified primary human NK cells at a 2:1 ratio (CHO:NK). Following this co-incubation, a Lipofect-amine/PolyI:C complex (100 µg/mL) was added to each well to stimulate a response for 24 hr. To investigate the role of LFA-1, NK cells were pre-incubated with a neutralizing antibody (anti-CD11a, clone HI111) against the LFA-1 receptor for 30 min at RT before being added to the *Pf*GBP-130-CHO cells.

## siRNA-mediated knockdown of THP1 and NK cells

siRNA-mediated CD11a knockdown was performed in both THP-1 cells (ATCC) and primary human NK cells using Accell SmartPool siRNA (Dharmacon). THP-1 cells were seeded at $1 \times 10^5$ cells/well in a 96-well plate in Accell siRNA transfection medium containing 2.5% FBS and 1 µM CD11a-targeting siRNA. Primary NK cells ($1 \times 10^5$) were suspended in Accell siRNA transfection medium supplemented with 10 ng/mL IL-15 and transfected with 2 µM CD11a siRNA. Cells were incubated for 72 hr, after which knockdown efficiency was confirmed by immunoblotting with anti-CD11a antibodies.

## Flow cytometry

The following antibodies were used for staining: anti-human CD3 (biotin, OKT3; Elabsciences), anti-CD16 (3G8; BioLegend), anti-CD11a (HI111; BioLegend), CD56-PE (5.1H11; Elabsciences), Ultra-LEAF Purified Human IgG1 Isotype (BioLegend), Goat anti-Human IgG Alexa Fluor 488 (Invitrogen), Goat anti-Human IgG Secondary Antibody, Alexa Fluor 594 (Invitrogen), CD107a (H4A3; Elabsciences), CD25-Alexa Fluor 488 (BioLegend), and CD69 (FN50; Elabsciences). Purified NK cells, either alone or co-cultured with iRBCs, were stained in 100 µL PBS containing 0.2% BSA and 0.05% sodium azide for 30 min on ice. After washing, stained cells were analyzed on a BD LSR Fortessa X-20 flow cytometer, and data were processed using FlowJo software (BD Biosciences).

## NK cell co-culture with RBC and iRBC

Human NK cells were purified from PBMCs using the BioLegend Human NK Cell Isolation Kit, achieving >95% purity. Prior to co-culture with RBCs/iRBCs, NK cells were pre-incubated with Fcγ receptor blocker (anti-human CD16, clone 3G8; 2 µg per $10^6$ cells) for 30 min at RT. Late-stage *P. falciparum* schizont-iRBCs were purified at 0.5% parasitemia. For neutralization experiments, either anti-*Pf*GBP-130 antibody or isotype control IgG1 was added to the culture wells for 30 min prior to NK cell addition. Pre-treated NK cells were then co-cultured with iRBCs or uninfected RBCs at a 10:1 ratio (NK:iRBC or NK:RBC) for 48 hr. For activation marker analysis, cells were collected and stained with relevant antibodies. For parasitemia assessment, co-cultures were maintained for 96 hr. Cells were stained with Hoechst 33342 (20 µg/mL) and anti-human CD56-PE (BioLegend) for 15 min, washed, and analyzed using a BD LSR Fortessa X-20 flow cytometer.

## Statistical analysis

Data are presented as mean ± SEM. Differences between groups were analyzed via Student's *t* test. A p-value<0.05 was considered statistically significant. All calculations were performed using GraphPad Prism 10 software package.

## Acknowledgements

The research work in PMs and AMs is supported by the Flagship Project (BT/IC-06/003/91), DBT-Wellcome; Team Science Grant (IA/TSG/22/1/600422); and JC Bose Grant (DST/20/015) from the Department of Science and Technology, Govt. of India. The funders had no role in the design of the study; the collection, analysis, and interpretation of the data; or the writing of the manuscript. We also thank the Rotary Blood Bank (India) for providing human red blood cells. We also acknowledge the Immunobiology group at ICGEB for providing infrastructure and insightful suggestions for the completion of this study.

## Additional information

### Competing interests

Dhiraj Kumar: Reviewing editor, eLife. The other authors declare that no competing interests exist.

### Funding

| Funder | Grant reference number | Author |
| --- | --- | --- |
| Department of Science and Technology, Ministry of Science and Technology, India | BT/IC-06/003/91 | Pawan Malhotra |
| Wellcome Trust/DBT India Alliance | IA/TSG/22/1/600422 | Pawan Malhotra |
| Department of Science and Technology, Ministry of Science and Technology, India | DST/20/015 | Pawan Malhotra |

The funders had no role in study design, data collection and interpretation, or the decision to submit the work for publication. For the purpose of Open Access, the authors have applied a CC BY public copyright license to any Author Accepted Manuscript version arising from this submission.

### Author contributions

Osama Mukhtar, Conceptualization, Data curation, Formal analysis, Investigation, Methodology, Writing – original draft, Writing – review and editing; Ravi Dutt, Gourab Paul, Formal analysis, Investigation, Methodology; Ashutosh Panda, Data curation, Formal analysis, Writing – original draft, Writing – review and editing; Poonam Kumari, Software, Investigation, Methodology, Writing – review and editing; Suneet Shekhar Singh, Data curation, Software, Formal analysis, Methodology; Neha Prakash, Madiha Abbas, Data curation, Investigation, Methodology; Md Muzahidul Islam, Priya Arora, Data curation, Formal analysis, Methodology; Alma Tammour, Data curation, Methodology; Asif Mohmmed, Conceptualization, Resources, Supervision, Funding acquisition, Writing – review and editing; Dhiraj Kumar, Conceptualization, Resources, Data curation, Supervision, Funding acquisition, Project administration, Writing – review and editing; Pawan Malhotra, Conceptualization, Resources, Supervision, Funding acquisition, Visualization, Writing – original draft, Project administration, Writing – review and editing

### Author ORCIDs

Dhiraj Kumar  https://orcid.org/0000-0001-7578-2930
Pawan Malhotra  https://orcid.org/0000-0002-7384-6280

## Ethics

The study was conducted according to the guidelines of the Declaration of Helsinki and approved by the International Centre for Genetic Engineering and Biotechnology (ICGEB)'s Scientific Ethical Review Unit and the Institutional Animal Ethics Committee (ICGEB/IEAC/25092023/39.11). Written informed consent was obtained from all healthy adult volunteers prior to blood collection. All samples were anonymized before analysis.

Reviewer #1 (Public review): https://doi.org/10.7554/eLife.110942.3.sa1
Reviewer #2 (Public review): https://doi.org/10.7554/eLife.110942.3.sa2
Reviewer #3 (Public review): https://doi.org/10.7554/eLife.110942.3.sa3
Author response https://doi.org/10.7554/eLife.110942.3.sa4

# Additional files

## Supplementary files

Supplementary file 1. Table A. Table of MS/MS hits of the beads+hIgG control. Table B. Docking scores and binding energy calculated using various computational tools to assess the quality of the GBP-LFA1 docked complex.

MDAR checklist

## Data availability

All the required data has been supplemented with the manuscript.

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
