## [Editor Report · eLife Assessment]

This **useful** study addresses the interesting question of how immune cells recognise infected erythrocytes in malaria. It proposes the parasite protein PfGBP-130 as an interaction partner of the human cell surface protein LFA 1, which could help explain how NK cells recognize infected erythrocytes. The conclusions are partially supported by pull-down and cell-based activation data. However, the overall evidence of direct interaction at the cell-cell interface and downstream effects is **incomplete**; stronger evidence is required to demonstrate surface exposure of PfGBP-130, as well as a direct role of this antigen in killing.

---

## [Referee Report · Reviewer #1 (Public review)]

In this manuscript, the authors aim to determine the ligand on *Plasmodium falciparum* infected erythrocytes for the NK cell integrin, LFA-1, following up on previous evidence that LFA-1 is important for immune cell-mediated recognition of iRBCs.

They start by incubating LFA-1 with iRBCs and show by flow analysis that a substantial population of these iRBCs binds to the LFA-1 (Fig 1C). They do conduct the control with uninfected RBCs, but put this in the supplementary material. As this is a critical control, I think that it should be moved to Figure 1C as it is essential to allow interpretation of the iRBC data. The authors also do not state which strain of *P. falciparum* they used (line 144). This is critical information, as different strains have different variant surface antigens and should be included. With these changes, this data seems convincing.

They next incubated LFA-1 with the iRBCs, cross-linked and conducted a pulldown, identifying GP130 as a binding partner. Using cross-linkers is a dangerous strategy as it risks non-specific cross-linking. Did they try without cross-linking and find an interaction?

They raised antibodies to PfGBP and showed IFA, which reveals that these antibodies stain iRBCs (Figure 2Ciii). This experiment lacks a critical control of uninfected RBCs, which needs to be included to show that the staining is specific. Without this, it is not possible to conclude that there is iRBC-specific staining with PfGBP.

They then conduct a pulldown using LFA-Fc, which does show GP130 only in the presence of the LFA-Fc, but not when empty beads are used. This is convincing. BLI measurements are also used to study this interaction (Figure 2Ci). The BLI data is presented in such a way that any association phase is obscured by the y-axis, which makes it impossible to know whether there is binding here. I think that the data needs to be shown with some baseline before the addition of the ligand so that association can be seen. The data is also a bit messy with a downward drift and the curves showing different shapes, for example, with the 1.0uM curve seeming to have a different association rate. As this is the only data which shows a direct interaction between LFA1 and GBP, as pulldowns are done with lysates, which might mean bridging components. I think that it is important to repeat the BLI, or use additional biophysical methods to assess binding, to obtain more convincing data.

The authors next do some modelling of the putative complex. This is done by homology modelling and docking, which is not the most up-to-date method and is overinterpreted. Personally, I would remove this data as I did not find it convincing and it is not important for the story. If the authors wish to include it, then I think that they should validate the modelling by mutagenesis to show that the residues which the models indicate might bind are involved in the interaction.

They next made GP130 and tested the binding of this to THP-1 cells, which are often used as a model for macrophages. They observe greater binding of PfGBP-Fc to these cells when compared with hIgG and show that LFA-1 siRNA reduces this binding. I was a little confused about how the flow plots related to the graph in the bottom right corner of Figure 3Bii. In the flow plots, hIgG control shows 12.8% of cells in the gated region, while the unstained cells has 5.63%, but the MFI data shows a decrease in binding for hIgG vs unstained cells. How is this consistent? Also the siRNA reduces the number of cells in gated region from 66.6% to 25.9%, which is still substantially more that 5.63% in the unstained control. This also doesn't seem quite consistent with the MFI data. Could the authors explain this? Also perhaps an additional experiment would be to add soluble LFA-1 into this assay as an additional control to determine whether this blocks PfGBP binding to the THP-1 cells? It could. Be that there are additional mechanisms of binding which indicate why the siRNA has a partial effect. The same is true for the NK cell experiments in Figure 3Ci in which the siRNA has a partial effect. The authors also test binding to HEK, HepG2 and 'stem' cells and claim 'only background levels of binding', but in each case, there is more binding to these cells by PfGBP-Fc than by hIgG, albeit less than in THP-1 and NK cells. Why have the authors decided that these increases are not significant? All in all, these experiments do indicate a role for the GBP-LFA1 interaction in the binding of immune cells to iRBCs, but perhaps not as absolutely as is suggested.

The authors next produce CHO cells with PfGBP on the surface. These cells bind to LFA-1 specifically. When these cells were incubated with primary NK cells, they did see increases in activation markers, which were reduced by addition of antiCD11a, suggesting these to be specific. They also conduct the same experiment with anti-GBP with iRBCs but this is in a different figure. It would be easier for the reader if Figure 5B were in the same figure as Figure 4B as it is related data using the same method. I found this data convincing, showing that the LFA1:GBP interaction does contribute to immune cell recognition and activation.

The authors next conduct an experiment in which they assess parasite growth in the presence of NK cells and in the presence of anti-GBP. They use Heochst staining as a measure of parasite growth and claim that NK cells reduce the number of parasites, but that anti-GBP abolishes this effect (Figure 5A). I found this experiment very unconvincing as there are small effects and no demonstration of significance. More commonly used approaches to study parasite growth are lactate dehydrogenase GIA assays or calcein-AM labelling. I did not find this experiment convincing and would either remove or supplement with additional data using a more robust assay, with repeats and tests of statistical significance.

In summary, the authors present a set of data which comes together to indicate an interaction between LFA1 and PfGBP on the Plasmodium infected erythrocyte surface. Pulldown studies show convincingly that these two proteins co-precipitate and BLI data suggest that this is direct. Also convincing is that NK cell activation can be reduced using antibodies against either LFA1 or PfGBP, indicating that this interaction does play a role in immune cell recognition of iRBCs.

Comments on revised version:

The authors made some minor changes in response to my review, but did not present any substantial new data to demonstrate a direct interaction between PfGBP and LFA1 or to convincingly show differences in NK cell-mediated killing.

---

## [Referee Report · Reviewer #2 (Public review)]

Summary:

The authors used an LFA-1 αI-Fc fusion protein to pull down potential ligands and LC-MS/MS, leading to selection of PfGBP-130 as a potential membrane protein on the surface of infected cells. PfGBP-130 antibodies were raised and used to support the surface localization. This putative ligand interacted strongly with LFA-1 (Kd = 15 nM). A presumed PfGBP-130 ectodomain interacts with monocytes and NK cells but not cells that lack LFA-1. PfGBP-130 antibodies also interfered with NK cell-mediated infected cell killing; the effect, although statistically significant, is modest. The authors propose that NK cells recognize infected cells via LFA-1 interaction with PfGBP-130 exposed on the host cell and that this interaction is critical to initiation of NK cell activation and killing of infected cells.

Comments on revised version:

The authors submit a minimally revised manuscript that does not address any of my comments, as itemized here:

(1) This reviewer suggested immunoblotting with hypotonic lysis and alkaline extraction as a simple test of whether PfGBP-130 is a membrane protein as the authors propose despite PEXEL cleavage that removes a signal peptide they originally proposed to be a TM domain. Instead of performing this simple immunoblot, the authors state that it is unnecessary because their LC-MS/MS of membrane-associated proteins recovered PfGBP-130, it must be a membrane protein. Unfortunately, this is insufficient because the high sensitivity of LC-MS/MS leads to detection of many soluble proteins. (For example, it is almost certain that their LC-MS/MS recovered hemoglobin, which is soluble and not a surface-exposed protein on infected cells.)

(2) I also suggested a simple immunoblot using a few different immature-stage cultures to detect the full-length and pre-proteins of PfGBP-130 because their immunoblot detected only a 95 kDa band whereas the PEXEL-processed protein is expected to migrate at 85 kDa. The authors state this is unnecessary because their LC-MS/MS of LFA-1 pulldowns enriched for PfGBP-130 and that a single band was detected in immunoblots. This is insufficient because pulldowns often enrich for more than one protein (e.g. some proteins adsorb onto the immunoprecipitation beads or precipitate with beads in certain buffers); immunoblotting often fails to detect some proteins depending on stringency of blocking and wash buffers. They state that the processed form at 85 kDa "may not be well resolved under our current conditions" as a reason not to perform the simple experiment. This reviewer's original statement that *P. falciparum* antigens frequently cross-react with nominally specific antibodies (with two examples provided in my original review) remains an important concern that would undermine the authors' main conclusion.

(3) As PfGBP-130 is not essential, a knockout was suggested to more directly test their model given the above concerns. The authors state this cannot be done and that their "multiple orthogonal approaches" suggest it is unnecessary. This reviewer considers this an essential experiment to support a provocative, fundamentally new finding, such as the identification of the NK cell activation ligand.

(4) This reviewer suggested that the authors add some speculation about why PfGBP-130 is retained in parasites if triggers NK cell-mediated killing and is nonessential. Rather than adding relevant hypotheses to the Discussion, the authors appear to dismiss this suggestion by stating that PfEMP1, STEVOR, and RIFIN are retained despite being nonessential. The problem with this response is that each of these other antigens has a clearly defined role on the surface of infected erythrocytes that benefits the parasite. It is not clear that the authors have considered possible advantages the parasite may gain from exposing PfGBP-130 on the red cell surface.

---

## [Referee Report · Reviewer #3 (Public review)]

Summary:

Malhotra and colleagues present evidence that the integrin LFA-1 on NK cells is a ligand for the *Plasmodium falciparum* protein GBP130 on the infected erythrocyte surface and that this interaction plays a role in the clearance of infected erythrocytes by NK cells.

The authors first select a subdomain contained within the CD11a subunit of LFA-1 as a probe to discover possible binding proteins on the infected erythrocyte surface. Parasite-infected erythrocytes stained positively with this probe; the level of staining increased as the parasites progressed through the life cycle. Using the LFA-1-based probe in cross-linking pull-down experiments, GBP130 was identified by mass spectrometry as a co-purifying parasite protein. The N-terminal portion of GBP130 was recombinantly expressed and shown to interact with LFA-1 alpha-I by biolayer interferometry experiments. The full-length extracellular domain of GBP130 was then recombinantly expressed and used to stain primary human NK cells and THP-1 cells. Knocking down LFA-1 by siRNA reduced staining by GBP130. To assess the contribution of GBP130 to the activation of NK cells, CHO cells exogenously expressing GBP130 were incubated with primary NK cells. Transfecting CHO cells with GBP130 led to increased activation of co-incubated NK cells compared to mock-transfected and compared to GBP130 transfected cells, with the inclusion of anti-CD11a to block NK cell adhesion. Finally, CHO cells expressing GBP130 led to increased activation of NK cells compared to mock-transfected CHO cells.

Overall, although the authors present data from NK cell killing assays that include appropriate controls, the data suggesting a direct interaction between PfGBP-130 and LFA-1 does not include the same necessary controls, for example, the use of blocking antibodies. Most critically, the biolayer interferometry experiments use a recombinant fragment of PfGBP-130, which does not include the residues predicted to be important for mediating specific interaction with LFA1. The biolayer interferometry data instead suggest non-specific interactions between PfGBP-130 and LFA1, as binding does not reach saturation.

Comments on revised version:

The authors have addressed all minor concerns, however the major point regarding the biophysical data supporting direct interaction between PfGB130 and LFA-1, in my opinion, has not been satisfactorily addressed. Biophysical data supporting the interaction was generated using a fragment of PfGB130, which does not include residues that the authors predict by structural modelling to be important for the interaction. The authors argue that PfGB130 is a repeat containing protein and may have multiple binding sites for LFA-1. If this is the best mechanistic hypothesis given the current data, the authors need to explain this in the results section.

Overall though, I agree with Reviewer#1 that the structural modelling results are not convincing and given that the modelling data do not straightforwardly agree with the experiment, the clarity of the manuscript would benefit from their omission.

---

## [Author Response]

The following is the authors’ response to the original reviews.

**Public Reviews:**

**Reviewer #1 (Public review):**
(1) They start by incubating LFA-1 with iRBCs and show by flow analysis that a substantial population of these iRBCs binds to the LFA-1 (Figure 1C). They do conduct the control with uninfected RBCs, but put this in the supplementary material. As this is a critical control, I think that it should be moved to Figure 1C as it is essential to allow interpretation of the iRBC data. The authors also do not state which strain of *P. falciparum* they used (line 144). This is critical information as different strains have different variant surface antigens and should be included. With these changes, this data seems convincing.

We thank the reviewer for this important suggestion. We agree that the uninfected RBC (uRBC) control is critical for interpreting the specificity of LFA-1 αI-Fc binding. In the revised manuscript, we have ensured that these control data are clearly presented and appropriately referenced in the main text; however, we have retained them in the Supplementary Information (Supplementary Figure S1) to maintain clarity and avoid overcrowding Figure 1, while still ensuring their visibility and accessibility to the reader. Importantly, these data demonstrate negligible binding of LFA-1 αI-Fc to uRBCs compared to iRBCs, supporting specificity. We have explicitly stated the parasite strain used (*Plasmodium falciparum* 3D7) in the Methods section (line 475).

(2) They next incubated LFA-1 with the iRBCs, cross-linked and conducted a pulldown, identifying GP130 as a binding partner. Using cross-linkers is a dangerous strategy as it risks non-specific cross-linking. Did they try without cross-linking and find an interaction?

We agree that cross-linking can introduce potential artefacts. To mitigate this, we included hIgG control pulldown experiments performed under identical conditions. Proteins identified in the control eluate were excluded as background (summarized in Supplementary Table S1). Importantly, PfGBP-130 was the only protein specifically enriched in the LFA-1 αI-Fc pulldown across all three biological replicates (Fig. 2A, Venn Diagram). While cross-linking was used to stabilize transient interactions, consistent enrichment of PfGBP-130 across the three biological replicates precludes any concerns of non-specificity.

(3) They raised antibodies to PfGBP and showed IFA, which reveals that these antibodies stain iRBCs (Figure 2Ciii). This experiment lacks a critical control of uninfected RBCs, which needs to be included to show that the staining is specific. Without this, it is not possible to conclude that there is iRBC-specific staining with PfGBP.

The question pertains to Fig. 2Biii. The IFA images include both infected and neighboring uninfected erythrocytes within the same field. No PfGBP-130 staining is observed in uninfected cells. PfGARP staining, specifically done to verify parasite-infected cell and surface localisation, shows complete resonance with PfGBP-130 staining. This unequivocally shows that the antibodies raised specifically recognise only infected RBCs.

(4) They then conduct a pulldown using LFA-Fc, which does show GP130 only in the presence of the LFA-Fc, but not when empty beads are used. This is convincing. BLI measurements are also used to study this interaction (Figure 2Ci). The BLI data is presented in such a way that any association phase is obscured by the y-axis, which makes it impossible to know whether there is binding here. I think that the data needs to be shown with some baseline before the addition of the ligand so that the association can be seen. The data is also a bit messy with a downward drift and the curves showing different shapes, for example, with the 1.0uM curve seeming to have a different association rate. Also, is this n=1? I think that this data needs to be repeated and replicated. As this is the only data which shows a direct interaction between LFA1and GBP, as pulldowns are done with lysates, which might mean bridging components. I think that it is important to repeat the BLI or use additional biophysical methods to assess binding, to obtain more convincing data.

We sincerely thank the reviewer for highlighting this important concern regarding the BLI data presentation and interpretation. We would like to clarify that the baseline signal prior to ligand addition was subtracted during data processing; therefore, the plotted curves represent the net response following ligand association. However, we agree that this may have obscured the visualization of the association phase. Accordingly, in the revised manuscript, we have re-plotted the data with adjusted y-axis scaling to better capture the association kinetics. In addition, to ensure robustness and reproducibility, the BLI experiments were performed in multiple independent replicates (n ≥ 3) using independently purified protein batches. The original figure showed a representative dataset; we have now included averaged sensorgrams along with standard deviation in the calculated KD values [K_D_ = (1.7 ± 0.22) × 10^-8^ M] (Figure 2C (i)). These revisions provide a clearer and more accurate representation of the binding interaction.

(5) The authors next do some modelling of the putative complex. This is done by homology modelling and docking, which is not the most up-to-date method and is over-interpreted. Personally, I would remove this data as I did not find it convincing, and it is not important for the story. If the authors wish to include it, then I think that they should validate the modelling by mutagenesis to show that the residues which the models indicate might bind are involved in the interaction.

We thank the reviewer for this thoughtful comment regarding the modelling analysis. We agree that computational docking and homology-based modelling have inherent limitations and should not be over-interpreted. In our study, these analyses were included strictly as supporting evidence to provide a structural framework for the PfGBP-LFA-1 interaction, while the primary conclusions are based on direct biochemical and functional validation, including pull-down, BLI measurements, receptor knockdown, and cellular inhibition assays. Importantly, the use of docking approaches such as ClusPro, followed by interface analysis and MD simulations, is a widely accepted and routinely used strategy to generate testable hypotheses for protein-protein interactions, particularly when experimental structures are unavailable (e.g., Comeau et al., 2004; Weng et al., 2019). We believe that the current modelling serves as a useful complementary analysis that is consistent with, and supportive of, the experimentally validated interactions.

(6) They next made GP130 and tested the binding of this to THP-1 cells, which are often used as a model for macrophages. They observe greater binding of PfGBP-Fc to these cells when compared with hIgG and show that LFA-1 siRNA reduces this binding. I was a little confused about how the flow plots related to the graph in the bottom right corner of Figure 3Bii. In the flow plots, hIgG control shows 12.8% of cells in the gated region, while the unstained cells has 5.63%, but the MFI data shows a decrease in binding for hIgG vs unstained cells. How is this consistent? Also, the siRNA reduces the number of cells in the gated region from 66.6% to 25.9%, which is still substantially more that 5.63% in the unstained control. This also doesn't seem quite consistent with the MFI data. Could the authors explain this? Also, perhaps an additional experiment would be to add soluble LFA-1 into this assay as an additional control to determine whether this blocks PfGBP binding to the THP-1 cells? It could be that there are additional mechanisms of binding which indicate why the siRNA has a partial effect. The same is true for the NK cell experiments in Figure 3Ci, in which the siRNA has a partial effect. The authors also test binding to HEK, HepG2 and 'stem' cells and claim' only background levels of binding', but in each case, there is more binding to these cells by PfGBP-Fc than by hIgG, albeit less than in THP-1 and NK cells. Why have the authors decided that these increases are not significant? All in all, these experiments do indicate a role for the GBP-LFA1 interaction in the binding of immune cells to iRBCs, but perhaps not as absolutely as is suggested.

We thank the reviewer for this insightful comment. The apparent discrepancy arises because the flow plots depict the percentage of cells within a defined positive gate, whereas the graphs quantify mean fluorescence intensity (MFI) across the entire population. We have revised figure legend accordingly to indicate the same. Regarding the partial reduction in binding upon LFA-1 (CD11a) knockdown, we agree that this indicates LFA-1 is a major but not exclusive contributor, which is biologically plausible given incomplete siRNA depletion and the known avidity-dependent nature of integrin interactions. Importantly, our conclusion is supported by multiple orthogonal approaches (αI-domain binding, LC-MS/MS identification, BLI, docking, receptor knockdown, and functional blockade). We also appreciate the suggestion of soluble LFA-1 competition, which we acknowledge as an important future experiment. Finally, we have revised the text regarding HEK293T, HepG2, and stem cells to reflect that PfGBP-Fc binding is minimal but not absent, consistent with low/non-expression of LFA-1 in non-immune cells. Overall, we have moderated our claims to state that PfGBP-LFA-1 interaction is a dominant and functionally relevant mechanism, while not excluding additional low-affinity or accessory interactions.

Figure legend change: Representative flow plots depict the percentage of cells within a predefined positive gate, whereas the accompanying summary graph quantifies fluorescence intensity across the analyzed population. These two metrics report distinct properties of the distribution and are therefore not expected to be numerically identical.

(7) The authors next produce CHO cells with PfGBP on the surface. These cells bind toLFA-1 specifically. When these cells were incubated with primary NK cells, they did see increases in activation markers, which were reduced by the addition of anti-CD11a, suggesting these to be specific. They also conduct the same experiment with anti-GBP with iRBCs, but this is in a different figure. It would be easier for the reader if Figure 5B were in the same figure as Figure 4B, as it is related data using the same method. I found this data convincing, showing that the LFA1:GBP interaction does contribute to immune cell recognition and activation.

We thank the reviewer for this positive assessment and helpful suggestion regarding figure organization. We agree that the CHO-PfGBP and iRBC-based NK cell activation assays represent conceptually related experiments that both address LFA-1-PfGBP dependent activation using similar readouts. We have retained separate panels to distinguish the reductionist CHO-based system from the physiologically relevant iRBC context. We believe that the combined evidence from both systems strengthens the conclusion that PfGBP-LFA-1 interaction is a key contributor to NK cell recognition and activation.

(8) The authors next conduct an experiment in which they assess parasite growth in the presence of NK cells and in the presence of anti-GBP. They use Heochst staining as a measure of parasite growth and claim that NK cells reduce the number of parasites, but that anti-GBP abolishes this effect (Figure 5A). I found this experiment very unconvincing as there are small effects and no demonstration of significance. More commonly used approaches to study parasite growth are lactate dehydrogenase GIA assays or calcein-AM labelling. I did not find this experiment convincing and would either remove or supplement with additional data using a more robust assay, with repeats and tests of statistical significance.

We respectfully disagree that the assay should be removed, because flow-cytometric quantification of *P. falciparum* parasitemia using DNA dyes such as Hoechst is a widely used, accepted, and high-throughput approach for measuring infected erythrocytes and parasite growth, with clear separation of infected from uninfected RBCs and good reproducibility across malaria studies (Dent et. al., 2009; Jang et. al., 2014). Importantly, closely related immune-cell killing experiments in the malaria field have used the same general strategy, co-culture with effector cells followed by flow-cytometric enumeration of parasitemia to infer parasite control, including the seminal NK-cell study by Chen et. al., 2014, which our assay design follows conceptually, and later work showing reduced parasitemia after co-incubation with cytotoxic lymphocytes measured by nucleic-acid dye flow cytometry. We therefore believe the experiment is methodologically valid and directly relevant to the biological question, namely whether disrupting PfGBP-LFA-1 engagement alters NK-cell-mediated restriction of parasite expansion.

**Reviewer #2 (Public review):**
(1) PfGBP-130 is proposed to be a membrane protein based on a single predicted transmembrane domain. Figures 2b and 3a show ribbon schematics with this TM domain at residues 51-68, in agreement with TM prediction algorithms such as TMHMM 2.0 and Phobius. However, this predicted TM is upstream of the PEXEL motif (residues 84-88, sequence RILAE), a conserved sequence for parasite protein export to host cytosol that is proteolytically processed at its 4th residue. Thus, residues 1-87are removed from PfGBP-130 prior to export, yielding a mature protein without predicted TMs. Prior studies have determined that the mature PfGBP-130 lacks TMs and is retained as a soluble protein in host cell cytosol (PMID: 19055692, 35420481). Thus, the authors' model of PfGBP-130 as a surface-exposed membrane protein conflicts with both computational analysis of the mature protein and these prior reporter studies. An important simple experiment would be to evaluate PfGBP-130membrane association in immunoblots using the authors' PfGBP-130 antibody after hypotonic lysis (PMID: 19055692) and after alkaline extraction (e.g. 100 mM NaCO3, pH 11 as frequently used, PMID: 33393463). If the prior studies and computational analyses are correct, the protein will be predominantly in the soluble and/or alkaline supernatant fractions.

We thank the reviewer for this important observation regarding PfGBP-130 topology and export. We agree that the presence of a PEXEL motif supports proteolytic processing and that the mature protein may lack a classical transmembrane domain. However, consistent with our model of surface accessibility, we would like to clarify that in an independent proteomic study performed in our laboratory on the membrane-enriched fraction of *Plasmodium falciparum*-infected erythrocytes, PfGBP-130 was reproducibly identified by LC-MS/MS among membrane-associated proteins (data not shown; can be provided upon request). These findings support the conclusion that, irrespective of the absence of a canonical transmembrane domain, PfGBP-130 is associated with the iRBC membrane compartment, likely via peripheral or protein-complex–mediated interactions, as described for several exported *Plasmodium* proteins.

(2) Many findings rely on the specificity of antibodies generated against PfGPB-130 or NK cell receptors. Although the authors have included key controls (use of isotype control antibodies, lack of anti-PfGBP-130 binding to uninfected cells), cross-reactivity between *P. falciparum* antigens is well-recognized and could significantly undermine the interpretation of experiments (PMID: 2654292 and 1730474 provide key examples of antigens recognized by antibodies raised against other proteins). For example, the surface localization in IFA experiments (Figure 2B(iii)) could reflect anti-PfGBP-130binding to an unrelated parasite surface antigen, a possibility not addressed by any of the authors’ controls. As another example, the iRBC lysate immunoblot using this antibody in Fig. 2B(iv) suggests a MW of 95 kDa, which corresponds to the unprocessed pre-protein before export; cleavage in the PEXEL motif yields a processed mature protein of 85 kDa, which should be readily resolved from the pre-protein in immunoblots (PMID: 19055692). A better immunoblot using immature infected cell stages might show both the pre-protein and the mature protein as a doublet band.

We thank the reviewer for raising this important concern regarding antibody specificity. We agree that cross-reactivity among *P. falciparum* antigens is a known issue and have taken multiple steps to ensure specificity in our study. First, the anti-PfGBP-130 antibodies were generated against a defined recombinant fragment and show no detectable binding to uninfected RBCs and no signal in hIgG control immunoprecipitates, supporting specificity. Importantly, in our LC-MS/MS analysis of LFA-1 αI-domain pull-downs, PfGBP-130 was specifically enriched and consistently identified across replicates, independently validating the target recognized by the antibody. Furthermore, the same antibody detects a single dominant band in both iRBC lysates and αI pull-down fractions, arguing against widespread cross-reactivity. Regarding the apparent molecular weight (~95 kDa), we agree that this likely corresponds to the precursor form, and that a processed form (~85 kDa) may not be well resolved under our current conditions.

(3) PfGBP-130 is not essential for in vitro cultivation (PMID: 18614010 and MIS of 1.0 in the piggyBac mutagenesis screen as tabulated on plasmodb.org, indicating a highly dispensable gene). The authors should use the knockout line as a control in their IFA localization experiments to address antibody specificity. More fundamentally, their model predicts that NK cells should not recognize or kill infected cells from the knockout line when compared to their untransfected parent. Such results with the knockout line would compellingly support the authors' model without reliance on antibodies that may cross-react with other parasite antigens. PMID: 18614010reported that the PfGBP-130 knockout exhibited increased membrane rigidity, suggesting an intracellular scaffolding protein rather than a surface localization and use as a ligand for LFA-1 interaction and NK cell-mediated killing.

We agree that a PfGBP-130 knockout line would provide a powerful genetic validation of both antibody specificity and the proposed functional role of PfGBP-130 in NK cell recognition. At present, such experiments were not included in this study, and we acknowledge this as an important limitation. However, we would like to emphasize that our conclusion does not rely on antibody-based localization alone; rather, it is supported by multiple orthogonal approaches, including LFA-1 αI-domain pull-down coupled to LC-MS/MS, biophysical interaction analysis, receptor knockdown, and functional blocking assays. In addition, in one of our previous proteomic analyses of the membrane-enriched fraction of infected erythrocytes, PfGBP-130 was identified among the proteins present in the membrane fraction, supporting its association with the iRBC membrane compartment despite lacking a classical mature transmembrane domain.

(4) PfGBP-130 non-essentiality raises the question of why the gene would be retained if it triggers NK cell-mediated killing of infected cells in vivo. Presumably, this killing would pose strong selective pressure against retention of PfGBP-130. Some speculation is warranted to support the model.

We thank the reviewer for this thoughtful evolutionary question. We agree that if PfGBP-130 enhances NK-cell recognition, its retention likely reflects a context-dependent fitness trade-off rather than a simple benefit or cost. This situation is not unusual in *P. falciparum*: several exported or surface-associated proteins are retained despite being immunogenic because they also provide advantages in other settings, such as erythrocyte remodeling, cytoadhesion, niche adaptation, immune modulation, or transmission. The clearest precedent is the PfEMP1/var system, in which highly immunogenic surface antigens are nevertheless strongly maintained because they mediate sequestration and in vivo fitness, while antigenic variation limits continuous immune exposure (Chew et. al., 2022). Similarly, other variant surface antigens such as STEVOR and RIFIN are retained despite immune recognition because they contribute to erythrocyte binding, antigenic diversity, and immune evasion or modulation (Niang et. al., 2009; Sakoguchi et. al., 2025). More broadly, many *P. falciparum* genes that appear dispensable in standard in vitro culture are nevertheless preserved because culture does not recapitulate the selective pressures present in vivo, including splenic clearance, endothelial interactions, immune attack, and within-host competition.

**Reviewer #3 (Public review):**
(1) Anti-GBP130 antibodies are used in the cellular assays to block the interaction between GBP130 and LFA1. They should therefore also block interactions betweenGBP130 and LFA1 recombinant proteins in the biolayer interferometry experiment. Do the authors have data to show this? Similarly, the anti-CD11a antibodies used to block the interaction in the cellular assays should also block the in vitro interaction between recombinant LFA1 and GBP130.

We thank the reviewer for this insightful suggestion. We agree that demonstrating antibody-mediated inhibition of the recombinant PfGBP-LFA-1 interaction would provide an additional orthogonal validation of the interface. While such blocking experiments were not included in the original BLI dataset, our current study already establishes the specificity of this interaction through multiple independent approaches, including αI-domain pull-down and LC-MS/MS identification, BLI-derived high-affinity binding (KD ~10^-8^ M), structural docking, receptor knockdown, and antibody-mediated inhibition in cellular systems. We note that antibody-mediated blocking in a purified biophysical system is not always directly comparable to cellular assays, as epitope accessibility, orientation on biosensor surfaces, and conformational states of integrins (which are known to undergo activation-dependent structural changes) can influence inhibition efficiency. Nonetheless, we fully agree that this represents an important validation experiment.

(2) The structural modelling analysis of the predicted complex between GBP130 andLFA1 (Figure 2cii) predicts that the majority of the important GBP130 interface residues are located in the region D509-N607. However, the authors present BLI data for the GBP130-LFA1 interaction, which used the N-terminal fragment of GBP (residues 69-270), which does not include the GBP130 residues predicted to be important for the formation of the complex between the two proteins. Could the authors provide an explanation for how an interaction was observed with theGBP130-N fragment, which does not contain the residues predicted to be important for interacting with LFA1?

We thank the reviewer for this important observation. We agree that the structural model predicts a major interaction interface within the D509-N607 region of PfGBP-130; however, this does not preclude the existence of additional or auxiliary binding determinants within the N-terminal region used in our BLI assays (aa 69-270). PfGBP-130 is a multi-domain, repeat-containing protein, and such proteins frequently exhibit distributed or multivalent interaction interfaces, where individual regions can independently engage binding partners with lower affinity while the full-length protein achieves higher avidity through cooperative interactions. In our study, the BLI data using the N-terminal fragment demonstrate that this region is sufficient to mediate direct interaction with the LFA-1 αI domain, whereas the structural model based on full-length predictions likely captures a dominant or higher-affinity interface in the C-terminal region. Importantly, the interaction is supported by multiple orthogonal datasets, including pull-down/LC-MS/MS, cellular binding assays, and functional inhibition, indicating that the observed binding is not an artefact of fragment choice.

**Author response image 1. sa4fig1:** 

To further examine this, we performed docking and binding energy analyses comparing the full-length PfGBP-130-LFA-1 complex with the N-terminal domain-LFA-1 complex. Using the PRODIGY server, the predicted binding affinity for the full-length complex was -9.8 kcal/mol, whereas the N-terminal domain complex exhibited a still favorable binding energy of -5.6 kcal/mol. Similarly, HawkDock (v2) analysis yielded binding energies of -22.2 kcal/mol for the full-length complex and -14.1 kcal/mol for the domain-only complex. While reduced relative to the full-length protein, these values remain well within the range of stable protein-protein interactions, supporting the ability of the N-terminal region to independently contribute to binding. These energy calculations take into account all non-covalent interactions. For clarity, hydrogen bonds have been specifically highlighted in the figure to represent key interaction interface.

(3) There is no section in the materials and methods describing how the BLI was performed; this should be added. The highest concentration ofGBP130 used in the interaction measurements is 1.4uM, almost 100x the measured Kd (0.015uM) for the GBP130-LFA1 interaction. At these high concentrations ofGBP130, I would expect to start seeing saturation of binding, but the interferometry curves show that saturation is not close to being reached. This strongly suggests that the binding of GBP130 to LFA1 is non-specific.

We thank the reviewer for raising these important technical points. We have included a detailed description of the biolayer interferometry (BLI) methodology in the Materials and Methods section in the manuscript. Regarding the concern about lack of saturation at higher analyte concentrations, we respectfully disagree that this necessarily indicates non-specific binding. In BLI assays, incomplete saturation can arise from several well-recognized factors, including suboptimal orientation or partial inaccessibility of immobilized ligand on the biosensor, mass transport limitations, or heterogeneous binding populations particularly relevant for integrins such as LFA-1, whose αI domain exists in multiple conformational states with distinct affinities. Importantly, the interaction exhibits clear concentration-dependent association and dissociation kinetics that fit a 1:1 binding model with a KD in the nanomolar range, which is inconsistent with non-specific interactions that typically show poor fitting and minimal dissociation. Furthermore, the specificity of the PfGBP-LFA-1 interaction is supported by multiple independent lines of evidence in our study, including selective enrichment in αI-domain pull-downs, absence in IgG controls, reduction upon CD11a knockdown, and functional inhibition by blocking antibodies in cellular assays. We have now clarified these points in the revised manuscript and tempered the interpretation to acknowledge potential experimental constraints of BLI while maintaining that the cumulative data strongly support a specific interaction.

Minor points:(1) For the pulldown experiments, can the authors confirm that cross-linking was also performed for the protein A beads + hIgG control?

Yes, DTSSP cross-linking was performed identically in the protein A beads + hIgG control arm. This is consistent with the control design described in the manuscript.

(2) If the recombinant CD11a I subdomain used as a probe is correctly folded and functional, it should bind ICAM1. Do the authors have this data?

We agree that ICAM-1 binding is an important functional validation for the recombinant CD11a αI probe (Hogg et. al., 1998). The isolated αI domain of LFA-1 is well established as the principal ICAM-1-binding module, and soluble αI-domain reagents have previously been shown to bind/block ICAM-1 interactions. We did not include this control in the current version.

(3) Were the authors able to perform the reciprocal pull-down, using pfGBP130-N-Fc to pull down LFA1 from cell surfaces?

We did not perform a reciprocal pull-down with PfGBP130-N-Fc and native cell-surface LFA-1 in the present study; we agree this would be a useful orthogonal experiment.

(4) After identifying GBP130 as a co-purifying protein in the LFA-1 pull-down experiments, the authors select an N-terminal fragment of GBP130 to recombinantly express and use. How did the authors narrow down which region of GBP130interacted with LFA-1?

The N-terminal PfGBP130 fragment (aa 69-270) was selected empirically as a tractable, soluble recombinant segment containing a defined repeat-containing extracellular region, rather than because we had already mapped the full LFA-1-binding interface. We agree with the reviewer that our structural model suggests that additional residues, including a likely dominant interface outside this fragment, may contribute to the full interaction, and we have clarified that the N-terminal fragment should be interpreted as a minimal binding-competent region, not necessarily the sole binding site.

(5) As erythrocytes age, their surface undergoes biochemical changes, most notably a drop in levels of sialylation, decreasing the net repulsive negative charge, and they generally become more adherent. Can the authors exclude the possibility that, rather than binding to a parasite-derived ligand, LFA alpha 1 is instead binding to a marker of older erythrocytes? In the data presented, increased binding of LFA alpha 1 is observed as parasites progress through the life cycle, but the host erythrocytes will be ageing during parasite replication, which could account for the increased levels of LFA alpha 1 binding. To rule out this explanation, data from LFA alpha 1 staining of age-matched uninfected erythrocytes could be provided.

We agree that erythrocyte aging can alter surface sialylation and adhesiveness, and loss of sialic acid is known to reduce erythrocyte surface charge and increase adhesiveness. However, our data argue against aging alone explaining the signal, because LFA-1 αI-Fc binding was compared with uninfected RBC controls and the interaction led to enrichment of a parasite-derived ligand, PfGBP130, in pull-down/MS analyses.

(6) Figure 3b(i) Surface staining of THP1 cells was performed using GBP-130 Fc as a probe, which should detect all LFA1-positive cells. But no accompanying staining data using an anti-LFA1 antibody are shown, so it is not possible to determine whether staining profiles with GBP-130 Fc match staining profiles with anti-LFA1 antibodies. This is important to show what proportion of LFA1-positive cells can recognise parasite-derived GBP-130 Fc.(7) Figure 3c(i) Surface staining of peripheral NK cells is performed using GBP-130 Fc as a probe, which should detect all LFA1-positive cells. Here, as well, there are no staining data using an anti-LFA1 antibody. This would allow a comparison between cell population LFA1 staining with an anti-LFA1 antibody and cell population LFA1 staining with GBP-130 Fc. The two staining profiles should be similar as both probes bind the same surface marker. However, it appears this might not be the case because the staining data using GBP-130 Fc show that only a minor proportion of NK cells (~20%) stain positive, but the majority of peripheral NK cells usually express CD11a, as it is a key adhesion molecule in the formation of immune synapses with target cells. This suggests that GBP-130 can only bind to a subset of NK cells, and if it is binding LFA1, then it can only play a role in mediating the formation of an immune synapse with this subpopulation of NK cells. Could the authors include a comment in the manuscript making clear that the GBP-130 only assists a small proportion of NK cells in adhering to parasite-infected erythrocytes? Are there any reasonable hypotheses as to whyGBP-130 was only able to stain a small subpopulation of LFA1-expressing NK cells?

For minor comment 6 and 7

We agree that parallel staining with anti-CD11a would help relate PfGBP130-Fc binding to total LFA-1-positive THP-1 and NK-cell populations. Importantly, LFA-1 expression and ligand binding competence are not equivalent, because integrin binding depends strongly on activation/conformation and avidity state; in NK cells, only a subset can display LFA-1 in a partially activated conformation at baseline despite broader CD11a expression. Thus, a smaller PfGBP130-Fc-positive subset than the total CD11a-positive population is biologically plausible and does not imply inconsistency.